# Inferring pattern-driving intercellular flows from single-cell and spatial transcriptomics

Axel A. Almet [1,2], Yuan-Chen Tsai [3,4,5], Momoko Watanabe [3,4,5] & Qing Nie [1,2,6] ✉

From single-cell RNA-sequencing (scRNA-seq) and spatial transcriptomics (ST), one can extract high-dimensional gene expression patterns that can be described by intercellular communication networks or decoupled gene modules. These two descriptions of information flow are often assumed to occur independently. However, intercellular communication drives directed flows of information that are mediated by intracellular gene modules, in turn triggering outflows of other signals. Methodologies to describe such intercellular flows are lacking. We present FlowSig, a method that infers communication-driven intercellular flows from scRNA-seq or ST data using graphical causal modeling and conditional independence. We benchmark FlowSig using newly generated experimental cortical organoid data and synthetic data generated from mathematical modeling. We demonstrate FlowSig's utility by applying it to various studies, showing that FlowSig can capture stimulation-induced changes to paracrine signaling in pancreatic islets, demonstrate shifts in intercellular flows due to increasing COVID-19 severity and reconstruct morphogen-driven activator–inhibitor patterns in mouse embryogenesis.

Cells communicate through biochemical signaling to organize biological activities. Inflows of intercellular signals are processed through intracellular gene regulatory mechanisms involving transcription factors (TFs) and their downstream targets, which result in outflows of other signals. These spatiotemporal flows of 'cause and effect' drive every biological process. One famous example of an 'intercellular flow' is Wolpert's French Flag Problem[1], wherein a spatially propagating morphogen drives coordinated expression of multiple TFs, generating the eponymous 'flag'. Biological homeostasis is maintained by coordination between intercellular flows, which is perturbed in disease. Disentangling these intercellular flows is critical to understanding health and disease.

scRNA-seq and ST generate simultaneous measurements of 10,000–20,000 genes, yielding high-dimensional snapshots of gene expression in biological tissue. From these data, patterns can be extracted that vary along axes such as trajectory, disease status, space

and time. There are two primary categories of methods to extract such patterns. First, one can construct gene expression modules (GEMs), defined by gene sets such that intra-set expression is more correlated than is gene expression between sets[2–10]. Second, one can infer ligand–receptor interaction networks that facilitate intercellular communication directly from non-spatial scRNA-seq[11–13] or spatial data[13–15]. The interplay between both ligand–receptor interactions and GEMs drives intercellular flows across tissues, but there are few methods that can infer such flows. We aim to address this gap.

In studies by Sachs et al.[16] and Chen et al.[17], which are similar to this work, graphical causal modeling was used to learn dependencies directly from single-cell data. Sachs et al. inferred a signaling network from multi-perturbation flow cytometry data of phosphoproteins measured in CD8+ T cells. Chen et al. inferred person-specific networks between GEMs generated from bulk RNA-seq and scRNA-seq data sampled from head and neck squamous cell carcinoma tumors. There

[1]Department of Mathematics, University of California, Irvine, Irvine, CA, USA. [2]NSF-Simons Center for Multiscale Cell Fate Research, University of California, Irvine, Irvine, CA, USA. [3]Department of Anatomy & Neurobiology, University of California, Irvine, Irvine, CA, USA. [4]Sue & Bill Gross Stem Cell Research Center, University of California, Irvine, Irvine, CA, USA. [5]School of Medicine, University of California, Irvine, Irvine, CA, USA. [6]Department of Developmental and Cell Biology, University of California, Irvine, Irvine, CA, USA. ✉e-mail: qnie@uci.edu

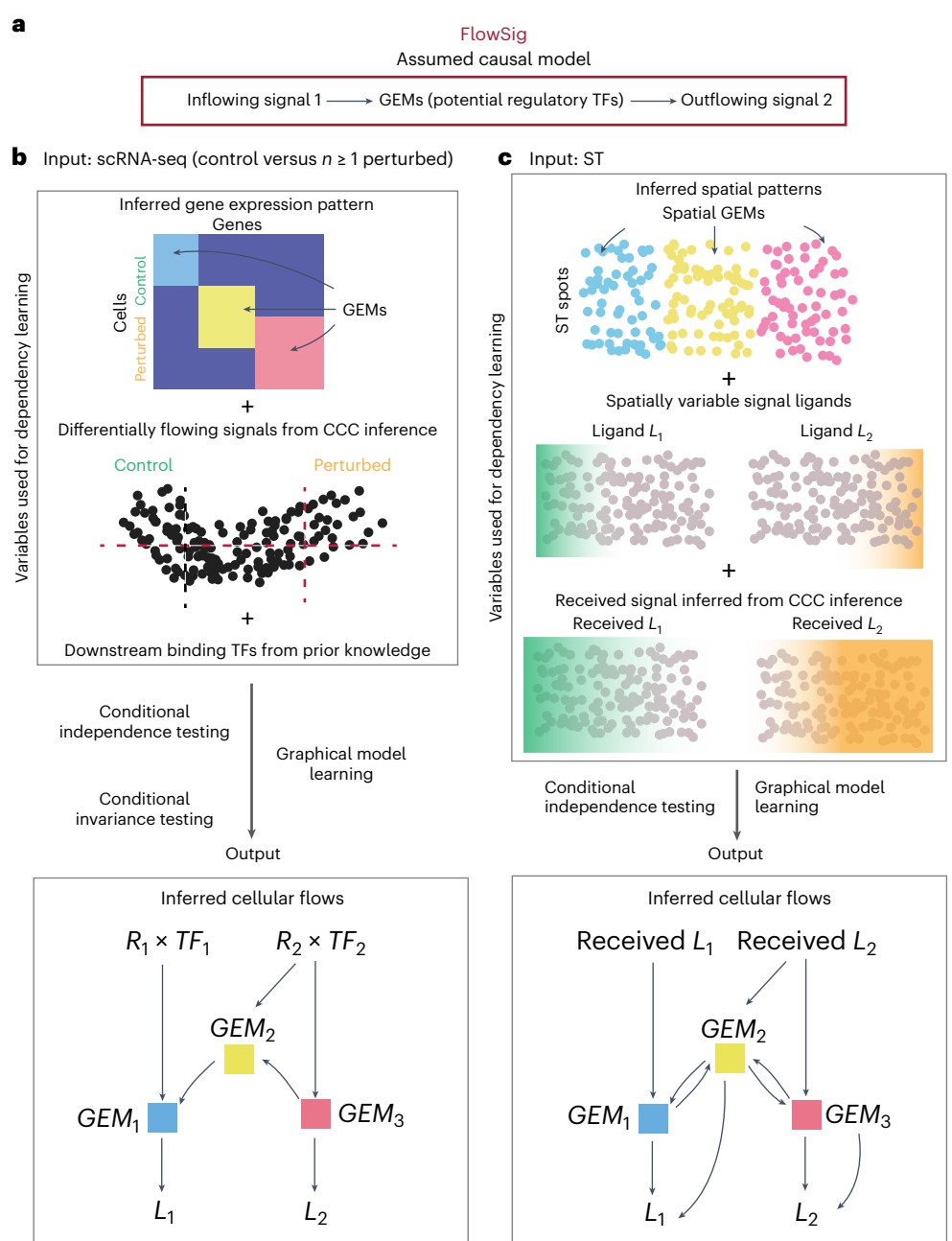

**Fig. 1 | Description of the FlowSig model. a,** We model intercellular flows to be directed from inflowing intercellular signals to GEMs that capture intracellular regulatory responses and that drive outflowing intercellular signals. FlowSig outputs an intercellular flow network describing directed edges from inflow signal variables (receptor gene expression weighted by the average expression of downstream TF gene set, $R_i \times TF_i$), to GEMs (cell membership to latent GEM factors, $GEM_i$) to outflow variables (signal ligand gene expression, $L_i$). **b,** FlowSig uses additional perturbation data and pathway knowledge of immediate downstream TF targets to learn accurate intercellular flows resulting from cell–cell communication. **c,** From spatial transcriptomics data, we can infer the amount of inflowing signals received at each spatial location more accurately, enabling us to infer intercellular flows without additional perturbation data. FlowSig outputs an intercellular flow network describing directed edges from inflow signal variables (inferred amount of received signal ligand from COMMOT, rec. $L_i$) to spatially resolved GEMs (membership to GEMs, $GEM_i$), to outflow variable (ligand gene expression, $L_i$).

is also the node-centric expression model by Fischer et al.[18] and the spatial variance component analysis framework by Arnol et al.[19], which infer how gene expression depends on the local environment. Other methods construct 'multicellular representations' of gene expression programs coordinated by several cell states[5,20–23] (see Supplementary Table 1 for a comparison of methods).

Here, we present FlowSig, a method that identifies ligand–receptor interactions whose inflows are mediated by intracellular processes and drive subsequent outflow of other intercellular signals. Using

graphical modeling and conditional independence testing, FlowSig learns a completed partial directed acyclic graph (CPDAG) describing intercellular flows between three types of constructed variables: inflowing signals, intracellular gene modules and outflowing signals. To reduce the false discovery rate, we orient the CPDAG according to the biological assumption that inflowing intercellular signals are processed by intracellular models before being converted to other outflowing signals. FlowSig can be applied to either non-spatial scRNA-seq or ST data. To analyze non-spatial scRNA-seq data,

in which ligand–receptor interactions are harder to infer accurately, we incorporate information gained from 'control versus perturbed' studies, in which the system has been altered by, for example, external stimulation, disease or time. FlowSig uses differential expression analysis and conditional invariance testing to infer the set of inflow and outflow variables that most significantly shift in distribution and thus most likely drive intercellular flows. In doing so, we reduce the set of possible graphs that could be generated by the data and learn a more accurate CPDAG. We validate FlowSig using (1) synthetic data generated from simulations of mathematical models of intercellular flows and (2) novel experimental data generated from cortical organoids. We benchmark FlowSig against several methods and show the unique insights gained from the platform. FlowSig is applied to scRNA-seq of stimulated human pancreatic islets, identifying specific changes due to stimulation. We analyze the case of multiple perturbations due to different COVID-19 severities resulting in distinct intercellular flow mechanisms. Applying FlowSig to ST data of mouse embryogenesis, we uncover regulatory TFs that enable a 'flow module' resembling Turing's activator–inhibitor system.

## Results

FlowSig uses gene expression measurements and output from cell–cell communication inference to learn intercellular flows that describe directed dependencies. These dependencies are oriented from inflowing intercellular signals to intracellular GEMs, which could be individual TFs or cellwise enrichment for correlated gene sets, and from GEMs to outflowing intercellular signals (Fig. 1a). We model the intercellular flows using graphical causal models, where nodes represent the flow variables—inflowing signals, GEMs and outflowing signals—and learn a directed graph using conditional independence testing and the unknown target interventional greedy sparsest permutation algorithm (UT-IGSP)[24]. Considering that one can use statistical conditional independence relations to infer, at best, a set of equivalent directed acyclic graphs (DAGs) with the same undirected skeleton graph and directed $v$-structures (connected node triplets $(x, y, z)$ with the directed edges $x \rightarrow y \leftarrow z$)[25], we use UT-IGSP to learn an initial CPDAG, which can contain both directed arcs and undirected edges. We then construct the intercellular flow network by reorienting undirected edges and removing biologically unrealistic arcs so that edges are directed from inflowing signals to GEMs, between GEMs and from GEMs to outflowing signals.

Although the core steps in using FlowSig to analyze non-spatial scRNA-seq and ST data are the same, there are several differences. For non-spatial scRNA-seq data, we must overcome a fundamental issue: it is not possible to directly measure the intercellular signals that each cell received. Therefore, we impose two constraints (Fig. 1b). First, we consider only studies comparing a 'control' condition against one or more perturbed conditions, for example, healthy versus diseased. We use the additional information gained from perturbation data through conditional invariance testing to narrow down the set of possible flow graphs, reducing the occurrence of false positive edge discovery. Second, for each ligand–receptor interaction inferred from cell–cell communication inference, we extract downstream TF targets from the OmniPath database[26] to measure signal inflow. Receptor gene expression quantifies the potential for a cell to receive an intercellular signal, and downstream TF expression quantifies the extent to which the cell actually received the signal; we define signal inflow as the product of receptor gene expression and the average expression of downstream TF targets.

ST technologies are currently in their infancy, so there are relatively fewer control versus perturbed ST studies than scRNA-seq studies. However, we can use communication methods such as COMMOT[14] to spatially constrain and measure the amount of inflowing signal more accurately (Fig. 1c). Therefore, FlowSig uses the greedy sparsest algorithm (GSP)[27], which does not use perturbation data, to analyze ST data.

### Synthetic validation of FlowSig

We first benchmarked FlowSig using synthetic data generated from mathematical models of intercellular flows (see 'Generating synthetic data from model simulations' in the Supplementary Notes). For simplicity, we modeled GEMs as individual TFs. We considered three scenarios. In the first scenario, we examined unidirectional intercellular flow induced by SHH signaling that generates outflow of BMP4 through FOXF1 (ref. 28), with flows learned over a set of five nodes: SHH ligand, unbound PTCH1 receptor, SHH inflow due to SHH–PTCH1 binding, FOXF1 TF and BMP4 ligand (Fig. 2a). The second scenario involved SHH-induced tissue patterning, characterized by the expression of NKX2.2, OLIG2, PAX6 and IRX3 (ref. 29). Flows were inferred over a set of seven nodes: SHH ligand, unbound PTCH1 receptor, SHH inflow (SHH–PTCH1 complex), NKX2.2 TF, OLIG2 TF, PAX6 TF and IRX3 TF (Fig. 2b). In the third scenario, we explored competition between SHH and BMP4 in driving dorsoventral patterning[30]. Flows were learned over a set of nine nodes, including SHH ligand, unbound PTCH1 receptor, inflowing SHH (SHH–PTCH1 complex), BMP4 ligand, unbound BMP1A and BMPR2 receptor, inflowing BMP4 (BMP complex) and three GEM variables, dorsal, intermediate and ventral (Fig. 2c). We wanted to validate two core FlowSig assumptions. The first is that accurate measurement of inflowing signal is needed to infer intercellular flows. For all models, we compared the use of bound ligand–receptor complex as signal inflow to total receptor expression (free receptor plus bound complex), the latter of which is directly measured from scRNA-seq and ST data. The second is that including perturbation data increases the accuracy of intercellular flow inference. We quantified the accuracy of FlowSig by measuring the true positive rate (TPR) and true negative rate (TNR) for each scenario. For all scenarios (Fig. 2d–f), we found that the average TPR does not change if we use bound receptor expression to measure signal inflow, or if perturbation data are introduced. However, measuring inflow using bound receptor increases the average TNR. This is especially true for the models describing SHH-driven patterning and competition between SHH and BMP4, in which flows are more complex and multidirectional (Fig. 2e,f). Incorporating perturbation data through conditional invariance testing reduces the variation in TNR values, both in terms of the interquartile range and outliers, resulting in 'tighter' estimates of intercellular flows. These results suggest that FlowSig reduces the number of false positive discoveries inferred from baseline GSP and UT-IGSP algorithms.

### Benchmarking FlowSig against multicellular representation methods

To provide additional insight into FlowSig's capabilities, we benchmarked it against methods that construct multicellular program representations from scRNA-seq and ST data, including DIALOGUE[5], scITD[20], MOFAcellular[22], MOFAtalk[22], MultiNicheNet[23] and Tensor-cell2cell[21]. We also compared FlowSig with direct CellChat output (Supplementary Table 1). All methods were benchmarked using an scRNA-seq dataset of stimulated peripheral blood mononuclear cells sampled from people with lupus, which was generated by Kang et al.[31]. We summarize key points here (see 'Comparison to other methods' in the Supplementary Results for a full discussion). We also evaluated FlowSig's robustness to different inputs constructed by alternative cell–cell communication and GEM construction methods (see 'Robustness of FlowSig to different input methodologies' in the Supplementary Results) and found that different cell–cell communication methods can result in different sets of intercellular flows, owing to discrepancies in inferred ligand–receptor interactions; however, FlowSig will infer intercellular flows through GEMs constructed by different methods that are enriched for the same regulatory TFs.

Analyzing CellChat output directly suggested there were 6,886 potential inflow-to-outflow relationships. Of these, 3,167 were shared across both conditions, 1,511 were unique to the control condition and 2,208 were unique to the stimulated condition. From CellChat results

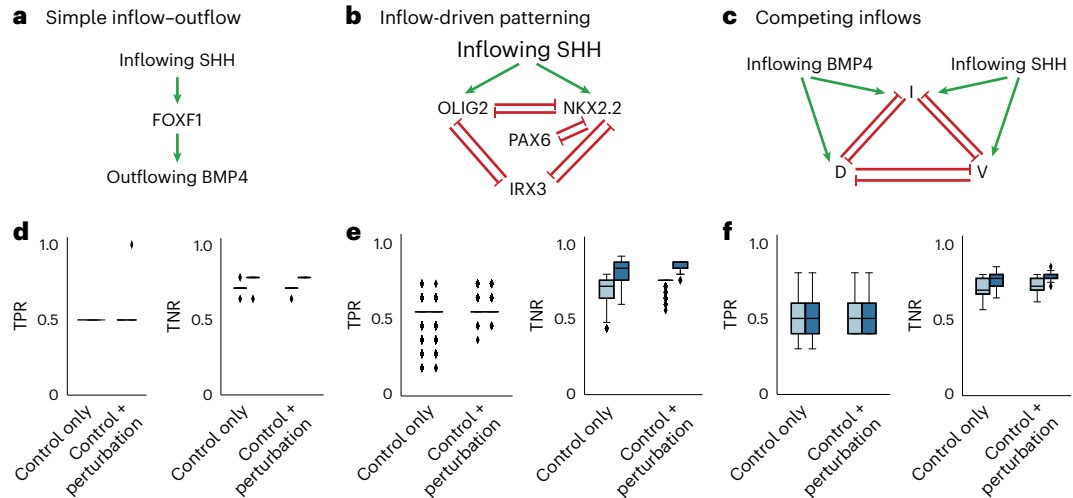

**Fig. 2 | Synthetic validation of FlowSig. a–c,** Causal diagrams representing activation (green arrow) or inhibition (red arrow) of unidirectional activation from inflowing SHH to outflowing BMP4 (**a**), SHH-inflow-driven patterning of NKX2.2, OLIG2, PAX6 and IRX3 (**b**) and competition between SHH inflow and BMP4 inflow to drive dorsoventral (dorsal (D), intermediate (I) and ventral (V)) patterning (**c**). **d–f,** The TPR and TNR of FlowSig output for **a–c**, respectively. We considered the effect of additional perturbation data and the effect of applying our biological flow model to constrain edges. In **d–f**, plots were generated over 500 simulations. Light blue boxes indicate the cases when total receptor expression (free plus bound receptor) was used as the inflow variable, while dark blue boxes indicate the cases when bound receptor expression was used as the inflow variable. Box plot whisker bounds are: **d**, minimum (TPR: control only 0.5, 0.5; control + perturbation: 0.5, 0.5; TNR: control only 0.56, 0.64; control + perturbation: 0.62, 0.72), maximum (TPR: control only 0.5, 0.5; control + perturbation: 0.5, 1.0; TNR: control only 0.79, 0.85; control + perturbation: 0.79, 0.85). Horizontal lines are defined by Q1 (TPR: control only 0.5, 0.5; control + perturbation: 0.5, 0.5; TNR: control only 0.56, 0.72; control + perturbation: 0.69, 0.77), median (TPR: control only 0.5, 0.5; control + perturbation: 0.5, 0.5; TNR: control only 0.69, 0.77; control + perturbation: 0.72, 0.77) and Q3 (TPR: control only 0.5, 0.5; control + perturbation: 0.5, 0.5; TNR:

control only 0.77, 0.79; control + perturbation: 0.77, 0.79). **e**, Minimum (TPR: control only 0.18, 0.18; control + perturbation: 0.36, 0.45; TNR: control only 0.56, 0.60; control + perturbation: 0.56, 0.76), maximum (TPR: control only 0.73, 0.73; control + perturbation: 0.73, 0.73; TNR: control only 0.80, 0.92; control + perturbation: 0.76, 0.88). Horizontal lines are defined by Q1 (TPR: control only 0.55, 0.55; control + perturbation: 0.55, 0.55; TNR: control only 0.64, 0.76; control + perturbation: 0.76, 0.84), median (TPR: control only 0.55, 0.55; control + perturbation: 0.55, 0.55; TNR: control only 0.72, 0.84; control + perturbation: 0.76, 0.88) and Q3 (TPR: control only 0.55, 0.55; control + perturbation: 0.55, 0.55; TNR: control only 0.76, 0.88; control + perturbation: 0.76, 0.88). **f**, Minimum (TPR: control only 0.3, 0.3; control + perturbation: 0.4, 0.4; TNR: control only 0.56, 0.64; control + perturbation: 0.62, 0.72), maximum (TPR: control only 0.8, 0.8; control + perturbation: 0.8, 0.8; TNR: control only 0.79, 0.85; control + perturbation: 0.79, 0.85). Horizontal lines are defined by Q1 (TPR: control only 0.4, 0.4; control + perturbation: 0.4, 0.4; TNR: control only 0.67, 0.72; control + perturbation: 0.69, 0.77), median (TPR: control only 0.5, 0.5; control + perturbation: 0.5, 0.5; TNR: control only 0.69, 0.77; control + perturbation: 0.72, 0.77) and Q3 (TPR: control only 0.6, 0.6; control + perturbation: 0.6, 0.6; TNR: control only 0.77, 0.79; control + perturbation: 0.77, 0.79). Diamonds indicate outliers (less than Q1 − 1.5 × IQR or greater than Q3 + 1.5 × IQR).

alone, we cannot infer which of these relations are truly intercellular flows, that is, whether the second interaction depends on the first, and we cannot infer the intracellular mediators of these intercellular flows. By contrast, FlowSig inferred only 44 intercellular flows across 6 signal inflow variables, 20 GEMs and 12 signal outflow variables (see 'Comparison to other methods' in the Supplementary Results and Supplementary Fig. 1).

DIALOGUE identified four multicellular programs (MCPs) from the Kang et al. dataset. MCP1 was enriched across CD14+ monocytes, CD8+ T cells and B cells, suggesting that there was coordination through intercellular flows between these cell types (Supplementary Fig. 2a). In MCP4, CD8+ T and CD14+ cells exhibited significant differential expression between conditions (Supplementary Fig. 2b). DIALOGUE identified upregulation of the signal ligand CCL4 (in CD8+ T cells), which FlowSig inferred to drive signal outflow. scITD decomposed the dataset into two latent factors (Supplementary Fig. 3a): Factor 1 was significantly enriched for FlowSig signal outflow ligands CXCL10, CXCL11 and TNFSF10 (Supplementary Fig. 3b) and intercellular-flow-driving interactions (Supplementary Fig. 3c). MOFAcellular decomposed the dataset into five factors (Supplementary Fig. 4a): Factor 1 was enriched for signal outflow variables CXCL11 and TNFSF10 (Supplementary Fig. 4b). Applying MOFAtalk to the ligand–receptor interaction scores inferred from LIANA[32] yielded four factors (Supplementary Fig. 4c): Factor 1 was enriched for the interactions CCL2–CCR1 and CCL8–CCR1 (between CD14+ cells, dendritic cells (DCs) and FGR3+ cells) and signal outflow of TNFSF13B (Supplementary Fig. 4d). Tensor-cell2cell

extracted six factors from ligand–receptor interaction scores inferred from LIANA (Supplementary Fig. 5a): CD14+ cells, DCs and FGR3+ cells were identified as key signal receiver groups (Supplementary Fig. 5b). Clustering ligand–receptor interactions identified that CCL2–CCR1, CCL3–CCR1, CCL4–CCR1 and CCL8–CCR1 were upregulated after stimulation (Supplementary Fig. 5c). Finally, MultiNicheNet identified CCL2–CCR1, CCL3–CCR1, CCL4–CCR1 and CCL8–CCR1 as differentially expressed between conditions (Supplementary Fig. 6a). MultiNicheNet also identified outflow of CXCL10, CXCL11 and FASLG and inflow into CCR1 (Supplementary Fig. 6b).

**Validating FlowSig using a cortical organoid system**

We tested FlowSig against new scRNA-seq data generated from an organoid model of cortical development, for which fibroblast growth factor (FGF) and bone morphogenetic protein (BMP) signaling are known to drive patterning[33]. We generated cortical organoids from human embryonic stem cells and collected the organoids at day 18 (D18) and D35 in culture for scRNA-seq analysis. In the organoid system, the cell fate for cortical identity is determined by D18, and signal responses to FGF and BMP, as measured by graded TF expression, are established by D35. The continual exposure of FGF and BMP signaling drives drastic changes in gene expression, and thus between D18 and D35 there are transcriptional changes and changes in cell type composition as the organoids mature. Hence, when applying FlowSig to this dataset, rather than assume the D18 and D35 populations are sampled from the same underlying 'steady state' distributions of gene transcription,

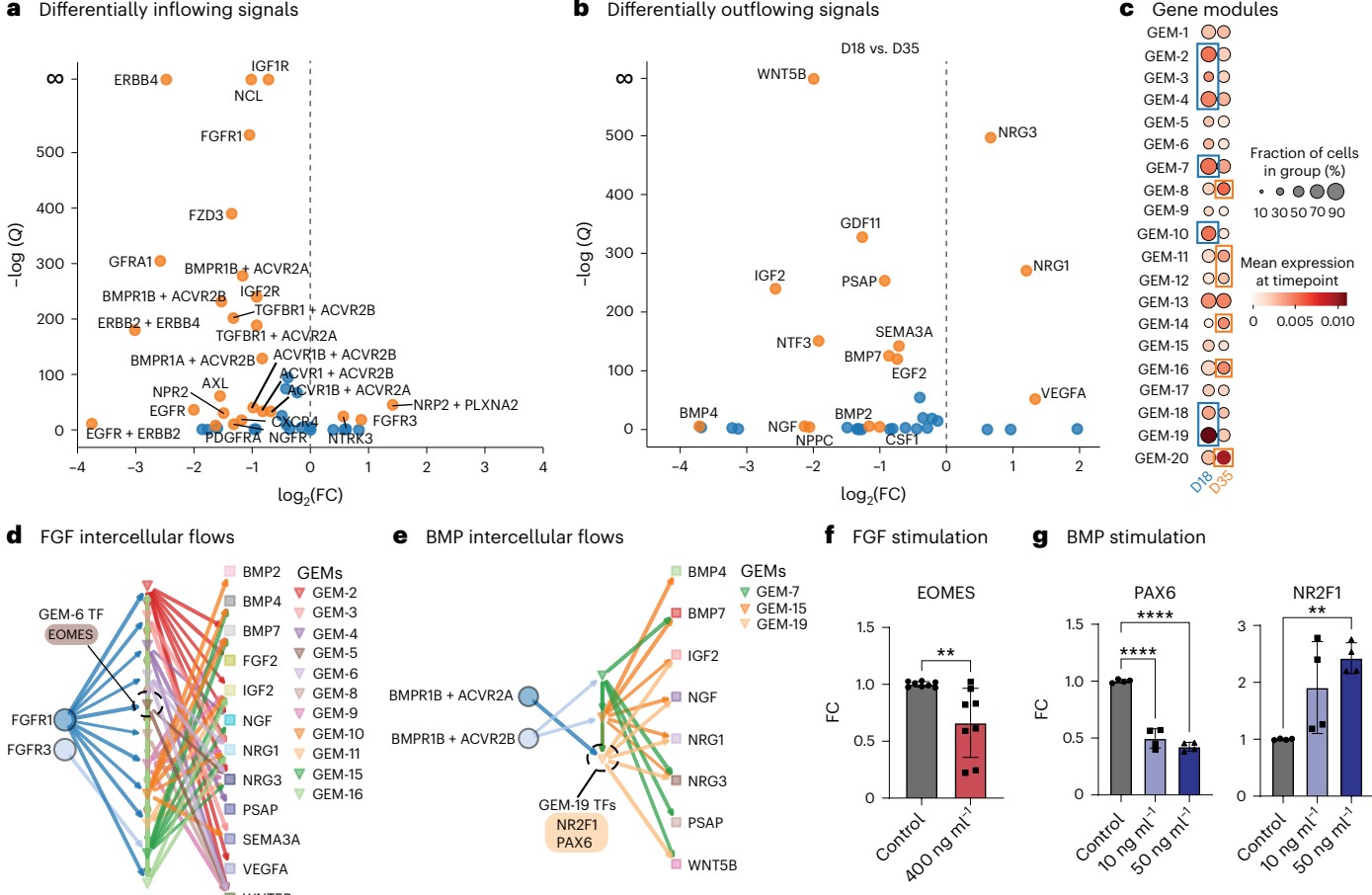

**Fig. 3 | Experimental validation of FlowSig using a cortical organoid model.**
**a**, Differentially inflowing signals between D18 and D35 timepoints.
**b**, Differentially outflowing signals between D18 and D35 timepoints.
**c**, Constructed gene expression modules capture time-specific and time-shared gene expression patterns. Some modules have been highlighted by the timepoint for which they are more enriched. **d**, Inferred intercellular flows due to FGF signaling. We speculated that EOMES is a key regulatory TF downstream of FGFR1 inflow. **e**, Inferred intercellular flows due to BMP signaling. We speculated that NR2F1 (CoupTF1) and PAX6 are downstream TF targets of BMP inflow.
**f**, Measurement of the FC of EOMES gene expression using RT–qPCR following bath application of FGF, with four biological replicates and two technical replicates. Unpaired two-tailed $t$-tests were performed ($t = 3.135$, d.f. = 14,

$P = 0.0073$). **g**, Measurement of the FC of PAX6 and NR2F1 (CoupTF1) gene expression using RT–qPCR following bath application of BMP, with two biological replicates and two technical replicates each. One-way ANOVA using Tukey's multiple-comparisons test was used to calculate adjusted $P$ values. For PAX6, control versus 10 ng ml$^{-1}$: mean diff. 0.50 with 95% confidence interval (CI) (0.40, 0.61), adjusted $P < 1 \times 10^{-4}$. For PAX6, control versus 50 ng ml$^{-1}$: mean diff. 0.57 with 95% CI (0.47, 0.68), adjusted $P < 1 \times 10^{-4}$. For NR2F1, control versus 10 ng ml$^{-1}$: mean diff. −0.91 with 95% CI (−1.88, 0.06), adjusted $P = 0.066$. For NR2F1 control versus 50 ng ml$^{-1}$: mean diff. −1.43 with 95% CI (−2.40, −0.45), adjusted $P = 0.0068$. In **f**, *$P \leq 0.05$, **$P \leq 0.01$, ***$P \leq 0.001$, ****$P \leq 0.0001$. In **g**, *adjusted $P \leq 0.05$, **adjusted $P \leq 0.01$, ***adjusted $P \leq 0.001$, ****adjusted $P \leq 0.0001$. Error bars represent s.d.

we treat the D35 data as a 'perturbed' form of the 'control' D18 data due to exposure to FGF and BMP signaling.

We identified differentially flowing signals from the 77 unique ligand–receptor interactions identified by CellChat[34] analysis. FlowSig identified 26 differentially inflowing signals (Fig. 3a) and 16 differentially outflowing signals (Fig. 3b), including FGF and BMP (see 'Identifying differentially flowing signal variables' in the Methods). We used pyLIGER[35] to construct 20 GEMs from 2,793 highly variable genes (Fig. 3c and Supplementary Fig. 8a–c). Cells from the D18 timepoint were more enriched for GEM-2 through GEM-4, GEM-7, GEM-10, GEM-18 and GEM-19, whereas cells from the D35 timepoint were enriched for GEM-8, GEM-11, GEM-12, GEM-16 and GEM-20. Altogether, FlowSig constructed 62 variables for intercellular flow inference. After inference, we aggregated signal inflow variables by their parent signaling pathway. For example, we classified both FGFR1 and FGFR3 inflows under the FGF signaling pathway, which were activated by received FGF2 ligand.

To determine the dominant drivers of intercellular flow, we ranked signal inflow variables by their total edge frequency. We found that FGF,

midkine (MK), pleiotrophin (PTN) and neuregulin (NRG) were drivers of intercellular flow. FGF inflow, in particular, drove signal outflow, including BMP4, insulin-like growth factor-II (IGF-II), nerve growth factor (NGF), NRG1 and NRG3, through numerous GEMs (Fig. 3d). By examining the top GEM-specific TFs mediated by FGF-induced flow (see 'Interpreting gene expression modules' in the Methods), we found that EOMES could be a potential regulatory candidate of FGF inflow. We observed that BMP inflow was regulated through many fewer GEMs (Fig. 3e) and could be mediated by PAX6 or NR2F1.

To verify FlowSig analysis, we analyzed a perturbed organoid culture in which we activated the FGF and BMP signaling pathways by adding FGF8b and BMP4, respectively, between D15 and D21. We collected organoid samples at D35 and subjected them to quantitative reverse transcription PCR (RT–qPCR) for gene expression analysis (Fig. 3f,g). Compared with the non-exposed control organoids, we observed that activating FGF signaling significantly downregulated the expression of EOMES (Fig. 3f), whereas elevating BMP signaling simultaneously downregulated PAX6 and upregulated NR2F1 (Fig. 3g).

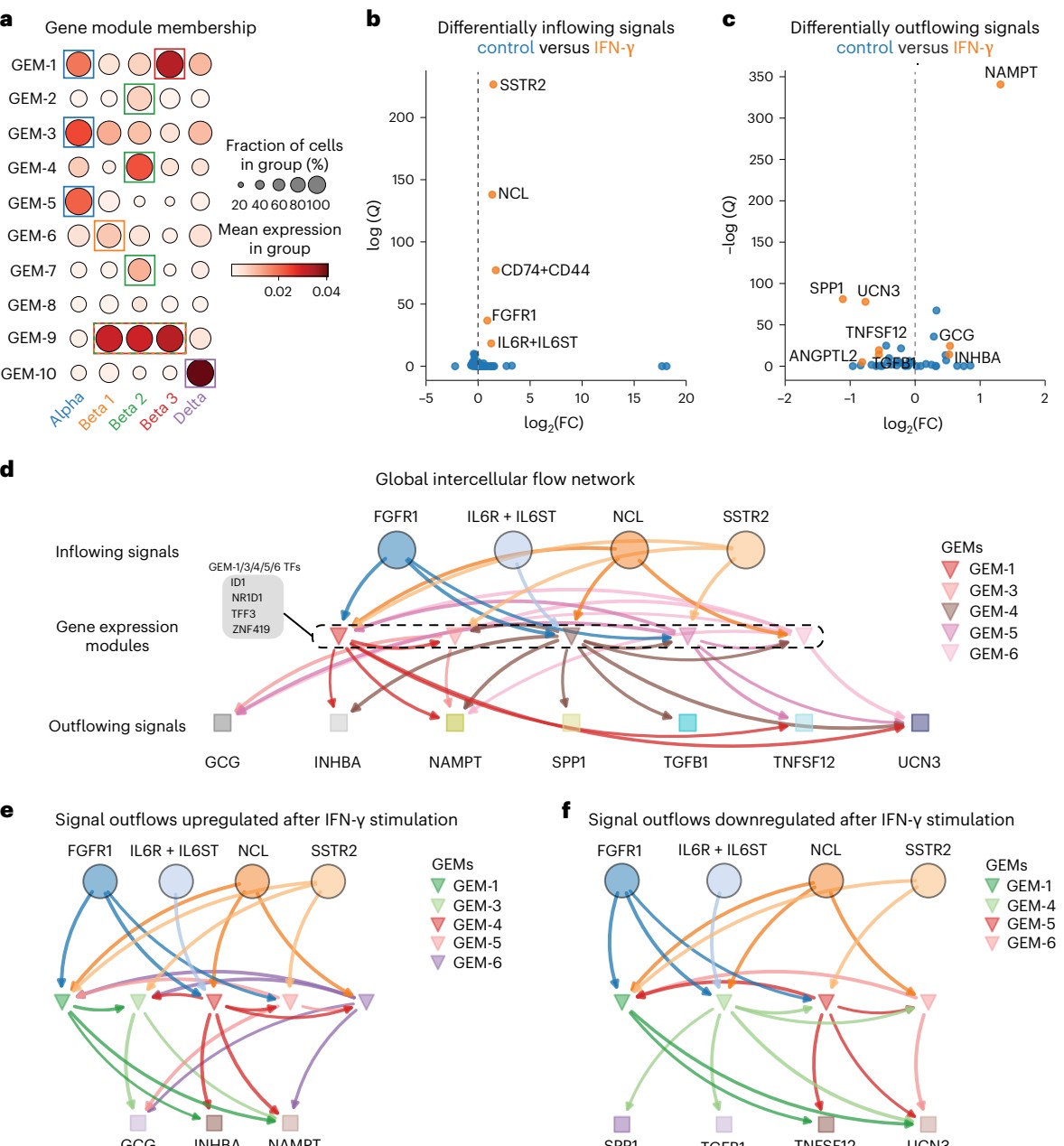

**Fig. 4 | Application of FlowSig to perturbed non-spatial scRNA-seq of pancreatic islets. a**, Constructed GEM modules align primarily with cell types identified from clustering. **b**,**c**, Differentially inflowing (**b**) and outflowing (**c**) signals due to IFN-γ stimulation. **d**, Identified global intercellular flow network inferred by FlowSig, capturing condition-shared and condition-specific flows. **e**,**f**, Intercellular flows regulating outflowing signals upregulated (**e**) and downregulated (**f**) by IFN-γ stimulation.

These experimental data demonstrate that FlowSig accurately captures the dominant drivers of intercellular flows from real biological datasets.

## FlowSig identifies changes in intercellular flows due to stimulation

To demonstrate how FlowSig recovers intercellular flows driven by an external perturbation, we analyzed scRNA-seq data of human pancreatic islets stimulated by interferon-γ (IFN-γ)[36]. We constructed ten GEMs using pyLIGER that aligned with the five cell-type clusters, alpha, beta 1 to 3, and delta, that we identified independently (Fig. 4a and Supplementary Fig. 9a–c). We used these cell type annotations as input for preliminary CellChat analysis; that is, for each condition, CellChat infers significant pairwise ligand–receptor interactions between the cell groups defined by these cell-type labels.

IFN-γ stimulation increased inflow of the FGF signaling pathway through FGFR1 (through ligands FGF7 and FGF9, specifically), interleukin-6 (IL-6) through IL-6R and IL-6ST, MIF through CD74 and CD44, MDK through NCL and SST through SSTR2 (Fig. 4b). IFN-γ stimulation increased outflow of GCG, INHBA and NAMPT, and decreased outflow of ANGPTL2, SPP1, transforming growth factor β1 (TGFβ1), tumor necrosis superfactor family member 12 (TNFSF12) and UCN3 (Fig. 4c). FlowSig identified that FGF, IL-6, MDK and SST were the dominant drivers of intercellular flows that drove the outflow of GCG, INHBA, NAMPT, SPP1, TGFB1, TNFSF12 and UCN3 through GEM-1, GEM-3, GEM-5 and GEM-6 (Fig. 4d). We observed that GEM-1 is enriched in both the alpha and beta 1 clusters, GEM-3 and GEM-5 are enriched in the alpha cluster, GEM-4 is enriched in the beta 2 cluster and GEM-6 is enriched in the beta 1 cluster (Fig. 4a), suggesting that

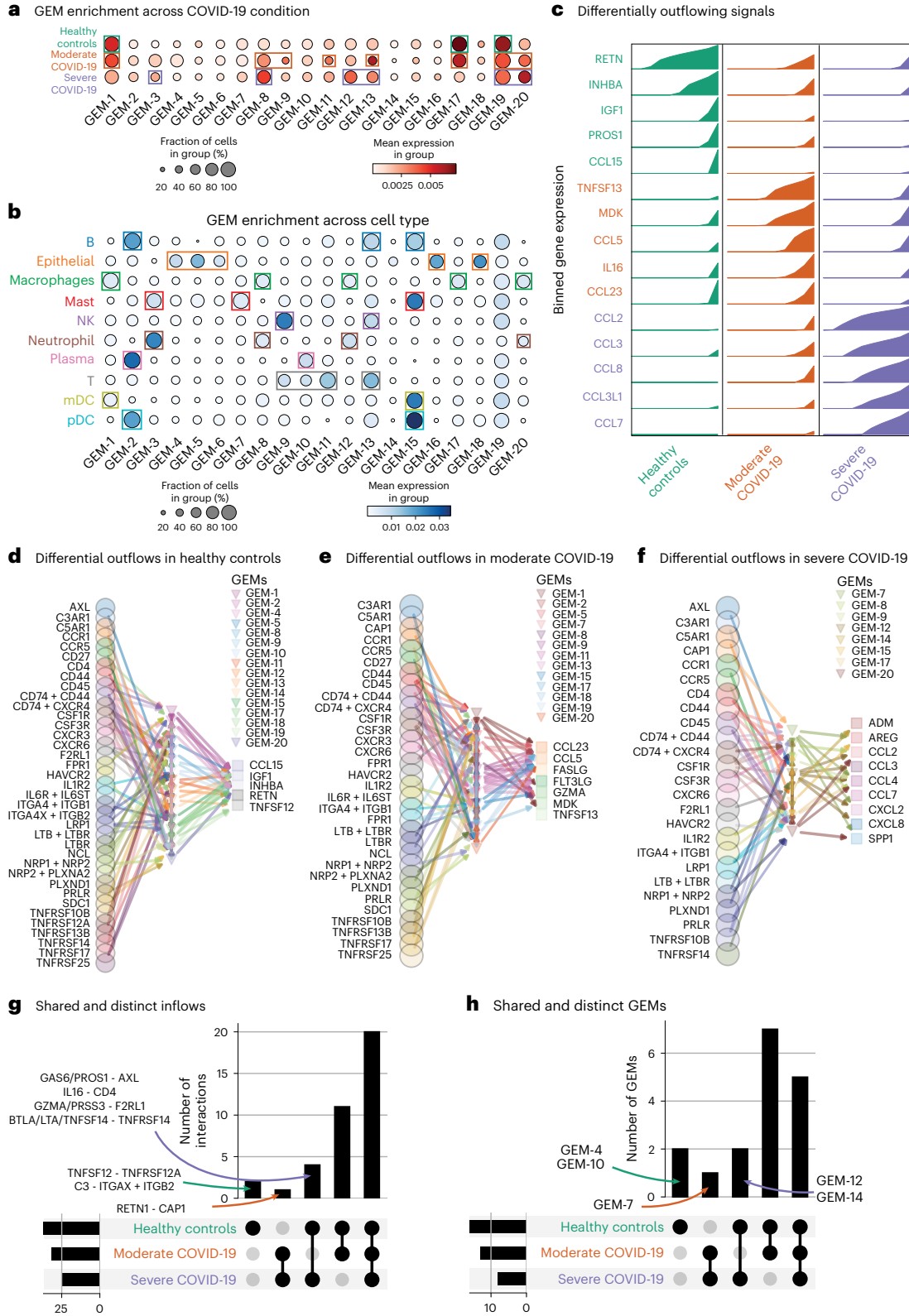

**Fig. 5 | Application of FlowSig to scRNA-seq of human BALF sampled from people with moderate or severe COVID-19. a,b,** Constructed GEM modules align with both COVID-19 conditions (**a**) and cell types identified from the original study (**b**). NK, natural killer; mDC, myeloid dendritic cell; pDC, plasmacytoid dendritic cell. **c,** Differential expression analysis identifies distinct sets of outflowing signal ligands for each condition. **d–f,** Identified intercellular flows driving outflow signals differentially expressed in healthy controls (**d**) and individuals with moderate (**e**) or severe (**f**) COVID-19. **g,h,** Upset plots quantifying the inferred inflow signals (**g**) and mediating GEMs (**h**) that are shared across COVID-19

conditions. The vertical bars represent the sizes of the intersection sets (number of ligand-receptor interactions and GEMs, respectively), and the horizontal bars represent the number of inflow signals (**g**) and GEMs (**h**) in each condition. In (**g**), some intersection sets are annotated with the constituent ligand-receptor interactions, of the form $L–R$, where $L$ is the ligand and $R$ is the receptor. In the case where multiple ligands competitively bind to the same receptor, different ligands are separated by a slash (/). For example, the interaction $L_1/L_2–R$ denotes that $L_1$ and $L_2$ are the ligands that can separately bind to the receptor, $R$.

intercellular flows are driven by cell type. These results agree with previous work establishing that, in the pancreas, alpha cells are the main secretors of GCG and beta cells are the main secretors of UCN3, and that SST regulates both GCG and UCN3 (ref. 37). We observed that the same TFs contributed to all of these GEMs—ID1, NR1D1, TFF3 and ZNF419—suggesting that these TFs mediate intercellular flows across both conditions.

To further explore the effects of IFN-γ stimulation, we split the global intercellular flow network into two networks. First, we constructed a network corresponding to outflow signals upregulated by IFN-γ stimulation by taking outflowing signals that were differentially expressed for the IFN-γ condition (adjusted $P < 0.05$ and log$_2$(fold change (FC)) > 0.5), the GEMs that connected to these outflow variables and the signal inflows nodes connected to these GEMs. From this node set, we then extracted the subgraph from the global intercellular flow network (Fig. 4e). The second network corresponded to the intercellular flow network of outflowing signals downregulated by IFN-γ (adjusted $P < 0.05$ and log$_2$(FC) < −0.5) and was constructed in a similar manner (Fig. 4f). Both networks contain the same signal inflow nodes and share near-identical GEMs. However, GEM-3, which drives GCG and NAMPT outflow and is itself regulated by SSTR2 (SST) signaling, is present only in the 'upregulated' network, suggesting that it has a specialized role activated by IFN-γ stimulation. GEM-3 is primarily enriched within alpha cells, suggesting that stimulation drives outflow of GCG and NAMPT from alpha cells. All other inflowing signals and GEMs are shared across both conditions, suggestive of dual regulatory roles. For example, IL-6 signaling drives both upregulation of INHBA and NAMPT and downregulation of SPP1, TGFB1 and UCN3 (through GEM-4).

## FlowSig uses multiple perturbations to find disease-driven changes

To demonstrate that FlowSig can handle multiple perturbations, we analyzed scRNA-seq of human bronchoalveolar lavage fluid (BALF) cells sampled from healthy controls and from people with either moderate or severe COVID-19 (ref. 38). We used CellChat and the cell-type annotations from the original study to infer significant ligand–receptor interactions, and found 46, 55 and 54 active signaling pathways for healthy controls and the moderate and severe COVID-19 groups, respectively.

We constructed 20 GEMs using pyLIGER (Supplementary Fig. 10a,b) that captured differences across both condition (Fig. 5a) and cell type (Fig. 5b). FlowSig identified differentially inflowing and outflowing signals specific to each COVID-19 condition with respect to healthy controls (Fig. 5c and Supplementary Fig. 10c,d). We note the differential expression of many inflammatory CC chemokines (CCLs) in severe COVID-19, including CCL2, CCL3, CCL8, CCL3L1 and CCL7, and CXC chemokines such as CXCL2 and CXCL8 (Supplementary Fig. 10d). In moderate COVID-19, we observed differential outflow of fewer inflammatory cytokines, including CCL5 and CCL23.

To analyze the intercellular flows driving these differential outflows, for each set of differentially outflowing signals, we extracted the upstream inflowing signals for which there was a directed path to at least one of the outflowing signals and the corresponding GEMs from the inferred FlowSig network (Fig. 5d–f). Despite the number of differentially outflowing signals increasing with COVID-19 severity, the number of inferred signal inflows decreased from 37 to 32 (loss of AXL, CD4, F2RL1, ITGAX and ITGB2, TNFRSF12A and TNFRSF14; gain of CAP1) to 25 (loss of CD27, CXCR3, FPR1, IL-6R and IL-6ST, LTBR, NCL, NRP2 and PLXNA2, SDC1, TNFRSF13B, TNFRSF17 and TNFRSF25; gain of AXL, CD4, F2RL1 and TNFRSF14). GEMs showed a similar trend: the number of regulatory GEMs decreased from 16 to 13 between healthy and moderate COVID-19 (loss of GEM-4, GEM-10, GEM-12 and GEM-14; gain of GEM-7). The results from Figure 5a,b suggest that the shift from healthy to moderate COVID-19 is associated with a downregulation

in intercellular flows through epithelial cells (GEM-4), plasma and T cells (GEM-10) and macrophages and neutrophils (GEM-12), but an upregulation of intercellular flows through mast cells (GEM-7). From moderate to severe COVID-19, there was a decrease from 13 to 8 (loss of GEM-1, GEM-2, GEM-5, GEM-11, GEM-13, GEM-18 and GEM-19; gain of GEM-12 and GEM-14).

We also calculated the intersections between the signal inflow sets (Fig. 5g) and GEM sets (Fig. 5h) driving signal outflows. We observed that 20 out of 37 signal inflows are shared across all three conditions. There were no signal inflows unique to either moderate or severe COVID-19 alone, whereas inflow through TNFRSF12A (due to TNFSF12) and ITGAX and ITGB2 (due to C3) drive outflows in only healthy controls. Only inflow through CAP (from RETN1) is shared between moderate and severe COVID-19 but is absent in healthy controls. There were more signal inflows shared between the healthy and moderate COVID-19 groups than between the healthy and severe COVID-19 groups or between the moderate and severe COVID-19 groups. We observed a similar trend amongst inferred regulatory GEMs. The most shared GEMs were between only the healthy and moderate COVID-19 groups (7 out of 17) and across all three conditions (5 out of 17). GEM-4 and GEM-10, which are associated with epithelial cells and T cells, respectively, mediated signal outflows in only healthy individuals. Only GEM-7, which is associated with mast cells, was shared between the moderate and severe COVID-19 groups but not healthy controls. No GEMs that regulate the differential outflows in severe COVID-19 were unique to the severe COVID-19 group. These results demonstrate how FlowSig can use multiple perturbations to identify trends in intercellular flows. Here, FlowSig identified that increasing severity of COVID-19 is associated with (1) a gradual loss of regulatory intercellular inflows and (2) an increase of inflammatory chemokine outflow that is driven by macrophages and neutrophils.

## FlowSig identifies regulators of spatial intercellular flow

We applied FlowSig to spatial Stereo-seq data of mouse embryogenesis sampled at stage E9.5 of embryogenesis[39]. We used non-negative spatial factorization[4] to construct 20 spatially resolved GEMs from 712 spatially variable genes (Fig. 6a and Supplementary Fig. 11a). We identified *Shh* outflow to be highly spatially variable (Moran's $I$ = 0.37; adjusted $P$ = 0.014; Supplementary Fig. 11b), and inferred *Shh* inflow across the tissue (Supplementary Fig. 11c), in line with *Shh*'s importance in development[40]. FlowSig identified several upstream drivers of *Shh* outflow, including *Bmp4*, *Cxcl12*, *Fgf15*, *Mdk*, *Ptn* and *Wnt5a*, which regulate *Shh* outflow through GEM-2, GEM-5, GEM-11 and GEM-14 (Fig. 6b) and inferred that received *Shh* inflow (denoted for brevity as *r-Shh*) drives outflow of several signal ligands through GEM-2, GEM-5, GEM-9, GEM-11, GEM-12, GEM-14, GEM-15 and GEM-17 (Fig. 6c).

We used these spatially resolved measurements to infer both specific upstream regulators of *Shh* outflow and downstream targets of *r-Shh* inflow. For each GEM, we extracted the top 10 TFs by module membership (see 'Interpreting gene expression modules' in the Methods). We identified potential upstream TFs of *Shh* outflow using random forest models[41], where we ranked TFs by feature (Gini) importance relative to all potential upstream TFs of *Shh* (see 'Inferring upstream TF regulators of spatial signals' in the Methods; Fig. 6d). We identified *Foxa2*, *Foxp2*, *Myc*, *Zc3h7a* and *Foxa1* as the top five upstream regulatory TFs of *Shh* outflow. Of these, *Foxa1* and *Foxa2* have been established to regulate *Shh*[42], as has *Foxp2* (ref. 43). Although *Myc* has been established to be regulated downstream of *Shh* signaling[44,45], its role as an upstream regulator is less clear.

To identify downstream targets of *r-Shh* inflow, we used pyGAM[46] (cubic splines, a gamma error distribution, and log link) to fit expression of the top 10 TFs of each inferred downstream GEM as a function of *r-Shh* inflow. We ranked TFs by the Spearman correlation between predicted TF expression and *r-Shh* itself (Fig. 6e). The downstream TFs that correlated least with *r-Shh* included known downstream targets

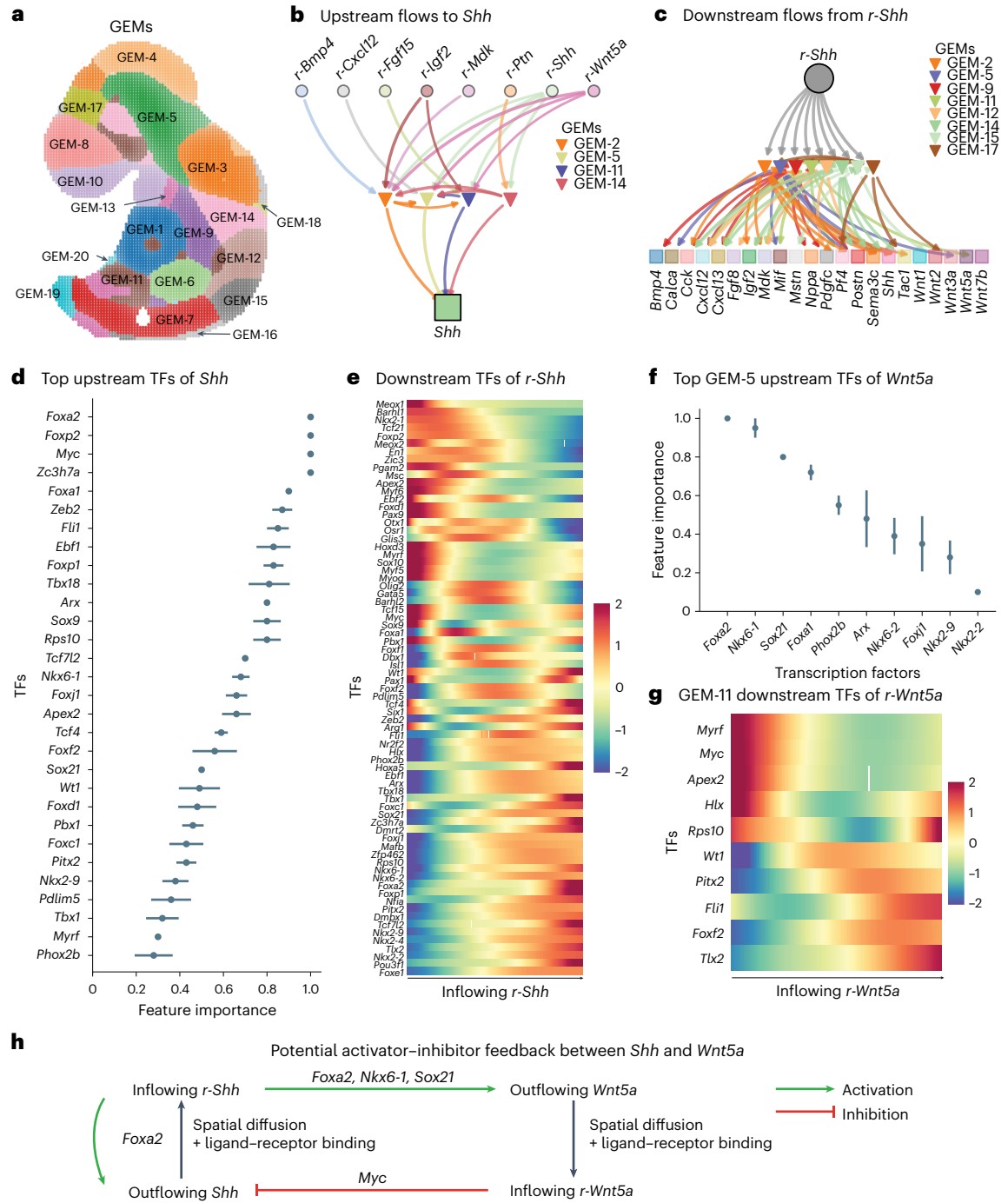

**Fig. 6 | Application of FlowSig to spatial Stereo-seq data of E9.5 mouse embryo. a**, Twenty identified spatial GEMs. Spatial spots are labeled by the GEM to which the spot membership is highest. **b**, Inflowing signals that drive *Shh* outflow. **c**, Identified downstream outflowing signals that are driven by inflowing *Shh* (*r-Shh*). **d**, Top upstream TFs ranked by their regulatory effects on outflowing *Shh*, as measured by random forest feature (Gini) importance. Feature importances were calculated over 10 runs. Data are shown as mean ± s.d. **e**, Potential downstream TFs regulated by inflowing *r-Shh*, ranked by Spearman correlation. The heatmap shows the scaled (*z*-transformed) values of the fitted gene expression values as a function of *r-Shh*. **f**, The top upstream TFs of outflowing *Wnt5a* that are also downstream targets of inflowing *r-Shh*. Feature importance was averaged over 10 runs. Data are shown as mean ± s.d. **g**, Potential downstream TFs regulated by *Wnt5a* that are also upstream regulators of outflowing *Shh*. **h**, Suggested activator–inhibitor feedback between *Shh* and *Wnt5a*, as implicated by **d**–**g**.

*Barhl1* (ref. 47) and *Nkx2-1* (ref. 48), as well as *Meox1*, *Tcf21* and *Foxp2*, whereas the TFs that were most correlated included known targets like *Foxe1* (ref. 49) and *Nkx2-2* (ref. 50), as well as *Pou3f1*, *Tlx2* and *Nkx2-4*. We observe that *Foxa2* is implicated both upstream and downstream of *Shh* outflow and inflow, respectively, suggesting that *Foxa2* could drive self-production of *Shh*.

We observed potential bidirectional flows between *Shh* and *Bmp4*, *Cxcl12*, *Igf2*, *Mdk* and *Wnt5a*. To validate these flows further, we performed the following analysis. For each ligand, we extracted the top GEM-specific TFs that were both upstream of the ligand and downstream of *r-Shh*. We used random forest modeling to calculate feature importance for each TF to ligand outflow. Only *Wnt5a* was

significantly regulated by TFs that were also downstream targets of *r-Shh* inflow through GEM-5 (Fig. 6f). Furthermore, outflowing *Wnt5a* and inflowing *r-Wnt5a* were spatially colocalized with inflowing *r-Shh* and outflowing *Shh* (Supplementary Fig. 11d,e). *Foxa2*, *Nkx6-1* and *Sox21* were the top upstream regulators of *Wnt5a* through GEM-5, in which *Foxa2* is known to regulate *Wnt5a*[51]. To infer whether inflowing *r-Wnt5a* regulated *Shh* outflow, we used pyGAM to fit the top TFs of GEM-11 as functions of *r-Wnt5a* inflow and ranked them by Spearman correlation of the predicted values with *r-Wnt5a* (Fig. 6g). We observed that *Myc*, one of the top upstream regulators of *Shh* outflow, negatively correlated with *r-Wnt5a* inflow.

These observations suggested the following bidirectional flow between *Shh* and *Wnt5a* (Fig. 6h). First, outflow and diffusion of *Shh* drives inflow of *r-Shh*, self-amplifying *Shh* outflow through *Foxa2*. Inflow of *r-Shh* also drives *Wnt5a* outflow through *Foxa2*, *Nkx6-1* and/or *Sox21*. Inflow of *r-Wnt5a* through spatial diffusion downregulates *Shh* outflow through *Myc*. This module resembles an activator–inhibitor system that can generate potential Turing patterns[52], with three key features. First, one or both signals can propagate—here, both *Shh* and *Wnt5a* ligands diffuse. Second, one of the signals—*Shh*—upregulates both itself through *Foxa2* and the other signal, *Wnt5a* through *Foxa1*, *Nkx6-1* and *Sox21*. Third, the other signal, *Wnt5a*, inhibits the activating signal, *Shh*. We found that *Wnt5a* inhibits *Shh* by downregulating *Myc*, an upstream regulator of *Shh*. It has been shown that activator–inhibitor systems can generate Turing patterns, which are defined by their complex spatial variation and are known to drive cell fate patterning in development[53–55], suggesting that at E9.5, *Shh* and *Wnt5a* play similar roles.

## Discussion

We developed FlowSig to infer intercellular communication activities that may depend on one another through coordinated GEMs. Key to our method is the construction of variables that measure either intercellular information (received and sent) or intracellular information. FlowSig applies graphical causal modeling and causal structure learning to scRNA-seq and ST data. As high-dimensional omics data continue to accumulate, the field will shift towards more predictive analyses, for which causal inference and causal structure learning models are likely to be key.

FlowSig complements the growing suite of methods for constructing multicellular representation programs. For example, DIALOGUE[5] uses multilevel modeling to extract coordinated programs involving two or more cell types that have significantly correlated gene expression. Such coordinated programs are likely mediated through the communication-driven intercellular flows that FlowSig can infer. Other methods, such as MOFAcellular[22] and scITD[20], decompose gene expression data into sample-specific and sample-shared latent GEMs that do not distinguish intercellular signal genes from intracellular signal-processing genes. MOFAtalk[22] and Tensor-cell2cell[21] extract coordinated programs of intercellular signaling from ligand–receptor interaction scores. Of the methods to which we compared FlowSig, the most similar is MultiNicheNet[23], which also constructs an intercellular signaling dependency network using pretrained signaling databases to construct the dataset-specific network; FlowSig uses conditional independence and conditional invariance testing to determine dependencies directly from the data.

To construct signal inflow and outflow variables, we used CellChat for non-spatial applications and COMMOT for spatial applications. There is a wide range of cell–cell communication inference methods[11,13], albeit with limited overlap in results[32]. Therefore, the choice of method can affect FlowSig output. Alternative communication methods, including CellPhoneDB[32] and LIANA[32], as well as alternative GEM construction methods, such as cNMF[7], can be used as input.

To reduce computation time, we inferred 'coarse-grained' intercellular flows, in which intracellular processing mechanisms are modeled

through multigene GEMs. We assume that these GEMs contain regulatory TFs that mediate signal inflow and outflow. Although we can extract downstream TFs from GEMs, we do not know the precise gene regulatory networks (GRNs) that mediate these signals. One could use methods such as SCENIC[56] to infer cellwise enrichment for significant regulons or incorporate data that measure open chromatin accessibility[57] to identify activated TFs. New data modalities, such as Phospho-seq[58], that measure post-translational response and thus signal inflow, will become useful for validation.

It is worth discussing FlowSig's limitations. As FlowSig uses conditional independence invariance testing based on partial correlation, the analyzed datasets must have sufficiently large sample sizes to estimate dependencies with sufficient statistical significance[59]. Furthermore, partial correlation assumes that the data are distributed according to a linear Gaussian model, which can be an unrealistic assumption[60]. Furthermore, as the number of variables increases, so too does the number of false positive relations inferred by the graph learning algorithms used by FlowSig. For non-spatial applications, to learn intercellular flows accurately, the perturbation must significantly shift the distribution of one or more variables. However, if the perturbation completely reduces signal variable expression to zero or induces expression of a variable not expressed in the control condition, partial correlation testing cannot be performed for the perturbed variable because it will have an s.d. of zero. One key limitation is that FlowSig infers a static graph, when intercellular flows are dynamic. Therefore, it will be important to extend FlowSig to capture spatiotemporal flows.

## Online content

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

## Methods

### FlowSig model

FlowSig's analyses are the same when applied to either non-spatial scRNA-seq or ST data. However, to compensate for the reduced precision of inflowing signals measurements from non-spatial scRNA-seq, we apply FlowSig to only scRNA-seq studies with an appropriate control condition and one or more perturbed conditions representing disease, external stimulation or biological time. We require input from intercellular communication inference and recommend using CellChat[61] and COMMOT[14] for non-spatial and spatial data, respectively. FlowSig provides functionality to construct GEMs from non-spatial data and NSF using pyLIGER. However, FlowSig flexibly allows users to use input from other cell–cell communication methods, such as CellPhoneDB[62] or LIANA[32], or from other GEM construction methods, such as cNMF[7]. We assume that, for each condition, the gene expression matrix ($X$) has been filtered and variance stabilized, for example by library-size normalization and log transformation. We note that original, unnormalized counts are also needed to construct GEMs. We use the input to construct augmented 'flow expression' matrices for each biological condition that measure inflowing signals, GEM enrichment and outflowing signals, which we define using three methods:

1. We define inflowing signals differently for non-spatial versus spatial data. For non-spatial scRNA-seq data, for each significant ligand–receptor interaction inferred from cell–cell communication analysis ($L$–$R$), we define the inflowing signal amount as $R \times \overline{TF}$, where $R$ is the receptor gene expression and $\overline{TF} = (TF_1 + \cdots + TF_m)/m$ is the average gene expression of the known immediate downstream TF targets that we infer from pathway knowledge databases, such as OmniPath[26] or exFINDER[63], where $m$ is the number of known TF targets (see 'Constructing downstream TF target sets to measure signal inflow' in the next section). For interactions involving receptor multi-units, $L$-$R_1 + \dots R_n$, where $n$ is the number of receptor sub-units, we use the geometric mean of receptor sub-unit gene expression values, $R = (R_1 \dots R_n)^{\frac{1}{n}}$, to calculate the inflow signal amount. Our rationale is that receptor gene expression quantifies a cell's 'potential' to receive intercellular signals, and the weighting by average downstream TF expression quantifies the actual downstream activation due to ligand–receptor binding and thus provides a more accurate measure of whether the cell actually received the signal. However, this definition is not exactly the same as the amount of 'received ligand,' which may not necessarily trigger downstream activation. By contrast, for ST data, we can measure the inflowing signal directly at each spatial spot using output from spatial CCC inference methods, such as COMMOT[14]. For a general method, for a given ligand ($L$) at ST spot ($S$), for every $L$–$R$ in which $L$ is inferred to partake, we define the inflowing signal amount as $\sum_R C_S^{(L-R)}$, where $C_S^{(L-R)}$ is the inferred communication score for interaction $L$–$R$ at spot $S$.

2. We defined GEM enrichment using output from matrix factorization methods, but GEM enrichment can be constructed from other dimensionality reduction methods in a similar manner. For matrix factorization methods, which decompose the gene expression matrix $X$ into $X = WH^T$ where, if $X$ is an $N \times G$ matrix, where $N$ is the number of cells and $G$ is the number of genes, $W$ is an $N \times K$ matrix describing cell membership into $K$ GEMs, where $K$ is the number of factors, and $H$ is a $G \times K$ matrix describing the loadings of each GEM, and $H^T$ is the transpose of matrix $H$, where the rows and columns have been interchanged to ensure correct matrix multiplication. Then, if we define $\bar{W}$ to be the normalized factor membership matrix such that the rows sum to unity, we define each GEM enrichment variable as $\bar{W}_k$, where $k = 1, \dots, K$. To standardize GEM enrichment values so that they are on the same scale as log-transformed gene expression

values, we use the log-transformed $\log(1 + \alpha\bar{W})$, where $\alpha$ is the scaling factor used to transform the original unnormalized counts, $Y = \log(1 + \alpha\bar{X})$, where $\bar{X}$ is the normalized gene expression matrix, such that the rows sum to unity.

3. Outflowing signals are defined as the gene expression of signal ligands implicated from cell–cell communication analysis. In the case of ligand multi-units, $L_1 + \dots L_n - R$, we use the geometric mean of ligand sub-unit gene expression values, $(L_1 \dots L_n)^{\frac{1}{n}}$.

Therefore, we associate cells with a vector containing three types of measurements: signal inflow measurements, which are receptor gene expression weighted by the average expression of their known downstream TF genes; intracellular 'module' enrichment, which is the cell's membership weight to a multigene set module, which measures how strongly the cell expresses those genes in the module; and signal outflow, which is ligand gene expression. When measuring signal inflow, we are not measuring from which cells the signals were sent, but rather how much signal has been received by the cell. Similarly, when measuring signal outflow, we are not measuring how much of the expressed signal ligand was actually received by other cells (as measured by, for example, signal inflow), but simply how much of the signal the cell is expressing.

FlowSig applies algorithms from causal structure learning that are based on the concepts of conditional independence testing and, if perturbation data are available, conditional invariance testing, to learn the directed intercellular flow network from the augmented flow expression matrices. Conditional independence testing infers the set of statistical dependencies from the data, whereas conditional invariance infers which variables shifted significantly in distribution after perturbation, for example, owing to disease or external stimulation. All conditional independence and conditional invariance testing are performed using partial Pearson's correlation to generate sufficient statistics. Despite partial correlation testing relying on the potentially unrealistic assumption that gene expression values are distributed according to a linear multivariate Gaussian distribution, we use the partial correlation method because it is significantly faster than other methods that use nonparametric kernel-based tests, and we can correct for biologically unrealistic edges by analyzing the learned CPDAGs rather than a DAG. To learn the CPDAG, we use the UT-IGSP[24] algorithm when analyzing non-spatial scRNA-seq with perturbation data and the GSP[27] algorithm for spatial data with no considered perturbation. Both of these methods estimate a CPDAG containing both directed and undirected edges that corresponds to the Markov equivalence class inferred from conditional independence and conditional invariance testing. Graphically, the Markov equivalence class is defined by the set of graphs that have the same skeleton graph, which is the undirected equivalent of the CPDAG, and $v$-structures, which are defined as directed node triplets $(x, y, z)$, where edges are oriented such that $x \rightarrow z \leftarrow y$. FlowSig reorients undirected edges inferred from UT-IGSP or GSP according to the assumption that inflow signal nodes must be directed towards GEM nodes, GEM nodes must be directed towards outflow signal nodes and edges between two GEM nodes can be bidirectional.

We also use bootstrap aggregation to further validate the learned intercellular flow network. For non-spatial scRNA-seq, we bootstrap by resampling individual cells from each condition with replacement. However, for ST data, we need to account for the spatial dependencies that affect correlation. Therefore, we perform a version of block bootstrapping[64] as follows. For each bootstrap realization, we divide the tissue into non-overlapping spatial regions, which we can obtain from either $k$-means clustering on the spatial coordinates, leiden clustering of the spatial connectivity graph or from predefined tissue region annotations. Then, within each 'block,' we resample with replacement. For each bootstrap realization, FlowSig outputs an adjacency matrix ($A$), that corresponds to the estimated CPDAG, where $A_{ij} = 1$ if an edge

has been inferred and $A_{ij} = 0$ otherwise. For $B$ bootstrap realizations, where $B > 0$ is the number of bootstrap samples, we then take the averaged adjacency, $\bar{A} = B^{-1} \sum_{b=1}^{B} A^{(b)}$, as the final CPDAG. To remove low-confidence edges, for every edge in the equivalent undirected skeleton graph of the CPDAG, we calculate the total edge weight as $w(i,j) = A_{ij} + A_{ji}$. For a specified threshold, defined by the parameter $w^* < 1$, if $w(i,j) < w^*$, we remove the edge from the network, that is, we set $A_{ij} = A_{ji} = 0$. Once the bootstrap aggregated CPDAG has been learned, biologically unrealistic arcs or edges are removed or reoriented, respectively. For all directed arcs from the filtered CPDAG, we retain only arcs that are directed from inflow signals to GEMs, GEMs to other GEMs or from GEMs to outflow signals. Similarly, for undirected edges, we orient edges such that nodes are directed in the same manner. In the case that an edge connects one GEM to another, we include both directions into the final intercellular flow network.

### Identifying differentially flowing signal variables

When inferring intercellular flows, we prioritize 'informative' inflowing and outflowing signal variables. In the case of scRNA-seq analysis, where perturbation data are available, we consider only 'differentially flowing' inflow and outflow signals. For all applications in this study, we use a Mann–Whitney $U$ (Wilcoxon rank-sum) test to assign variables as differentially flowing if their adjusted $P$ values (after correction for multiple hypothesis testing) fall below a specified threshold (for example, adjusted $P < 0.05$), indicating statistical significance, and whose $\log(FC)$ values are above a specified threshold (for example, $\log(FC) > 0.5$). We analyzed inflow signal variables separately from outflow signal variables. That is, we performed two separate Mann–Whitney $U$ tests—one to identify differentially inflowing variables from only the set of inflow signal variables and one to identify differentially outflowing variables from only the set of outflow signal variables. When analyzing ST data, in which perturbation data are not as readily available, FlowSig instead prioritizes inflow and outflow variables that are spatially variable. For all applications considered, we retain variables for which the graph-based global Moran's $I$, which we calculate using Squidpy[65], is above a specified threshold, for example ($I > 0.1$).

### Constructing downstream TF target sets to measure signal inflow

To measure signal inflow more accurately from non-spatial scRNA-seq data, we used prior knowledge from OmniPath[26] to weight the gene expression of receptors that have been implicated in intercellular communication from prior cell–cell communication inference. For each ligand–receptor interaction, we searched the KinaseExtra and PathwayExtra modules for TFs that are the first downstream targets of the relevant receptors. Because OmniPath has been constructed for human knowledge, when constructing the downstream TFs for mouse data, we convert the mouse receptor genes implicated from communication inference to their human orthologs and perform the same procedure as for human data.

### Interpreting gene expression modules

TFs are the mediators of signal transduction, that is, signal inflow, and the primary regulators of gene transcription, that is, signal outflow. To gain a deeper functional understanding of intercellular flows, it is important to interpret FlowSig output both with respect to GEMs, which describe the expression patterns of coordinated multigene sets, as well as individual GEM-specific TFs. For both non-spatial and spatial data, we consider only a priori known TFs, which in this case are based on TF lists provided by pySCENIC[56]. Specifically, we use the list provided in allTFs_mm.txt for mouse data and the list provided in allTFs_hg38.txt for human data.

For non-spatial scRNA-seq data, we used pyLIGER[35] to construct integrated GEMs. For a dataset describing $C$ conditions, pyLIGER uses joint matrix factorization to decompose each condition-specific gene expression counts matrix, $X^{(c)} \in \mathbb{Z}_{\geq 0}^{N \times G}$, where $\mathbb{Z}_{\geq 0}$ is the set of all

nonnegative integers, $N$ is the number of cells and $G$ is the number of genes, into $K$ GEMs through $X^{(c)} = F^{(c)} \cdot (W + V^{(c)})^{T}$, where $A^{T}$ is the transpose of matrix $A$, where rows and columns have been swapped. Here, $F^{(c)} \in \mathbb{R}_{\geq 0}^{N \times K}$ is the condition-specific factors matrix, describing the membership of the cells in condition $c$ to each of the $K$ GEMs, and $W \in \mathbb{R}_{\geq 0}^{G \times K}$ and $V^{(c)} \in \mathbb{R}_{\geq 0}^{G \times K}$ are the condition-shared and condition-specific loadings matrix, describing the membership of genes to each of the $K$ GEMs. Larger values of $F_{nk}^{(c)}$ correspond to greater membership of cell $n$ in condition $c$ to GEM $k$, while larger values of $W_{gk} + V_{gk}^{(c)}$ correspond to greater overall membership of gene $g$ to GEM $k$. We use the columns of $F^{(c)}$ as our $K$ GEM variables and use the columns of $W + V^{(c)}$ to extract the top TFs for each GEM. For each module $k$, we sort genes by decreasing order of the loadings sum, $W_{gk} + V_{gk}^{(c)}$, and then extract the top contributing TFs in the order by which they appear in the sorted lists.

For ST data, we use NSF[4] to construct spatially resolved GEMs. In brief, NSF decomposes the gene expression counts, $X \in \mathbb{Z}_{\geq 0}^{N \times G}$, which has $N$ spots and $G$ genes, into $K$ GEMs (factors) through $X = FW^{T}$, where the factors matrix, $F \in \mathbb{R}_{\geq 0}^{N \times K}$, describes the spotwise membership to the $K$ GEMs (factors) and is fit using Gaussian processes whose means and covariances vary with spatial locations. The loadings matrix, $W \in \mathbb{R}_{\geq 0}^{G \times K}$, describes the gene weight membership to each of the $K$ GEMs. Larger values of $F_{nk}$ indicate a higher enrichment of spot $n$ for GEM $k$, which describes a spatially varying gene expression pattern; larger values of $W_{gk}$ indicate greater membership of gene $g$ to GEM $k$, that is, how much gene $g$ contributes to the gene expression pattern. We use the columns of the factor matrix, $F$, as our $K$ GEM variables and use the columns of loadings matrix, $W$, to extract the top contributing TFs for each spatial GEM. For each module $k$, we sort all genes by decreasing order of their $W_{gk}$ value. We then extract the top contributing TFs by the order in which they appeared in the sorted list.

### Inferring upstream TF regulators of spatial signals

To infer which TFs could potentially regulate inferred signal outflow variables, we borrow the approach of Cang et al.[14] After FlowSig infers the global intercellular flow network, for each signal outflow variable that is connected in the network, we first backtrack through the directed network to infer which spatial GEMs are connected to the signal outflow node. For each GEM with a directed edge to the signal outflow variable, we extract the top 10 contributing TFs (see 'Interpreting gene expression modules' in Methods). We then use the scikit-learn implementation of the Random Forest regression model[66] to model the signal ligand gene expression as a function of the TF genes as independent variables. We then ranked the TFs with respect to their feature importance, which is calculated from the Gini importance (mean decrease in impurity).

### Experimental validation

**Human cortical organoid generation.** All experiments using human embryonic stem cells (hESCs) were approved by the University of California, Irvine (UCI) Human Stem Cell Research Oversight (hSCRO) Committee. The hESC line H9 was obtained from the WiCell Institute under a material-transfer agreement. The methods for hESC maintenance and cortical organoid production were previously established[67,68]. In brief, H9 cells were maintained with inactivated mouse embryonic feeders (PMEF-CF, Millipore Sigma) on a 0.1% gelatin-coated plate and cultured in DMEM/F12 (HyClone) with 20% knockout serum replacement (KSR, Invitrogen), non-essential amino acids (NEAAs, Invitrogen), GlutaMAX (Invitrogen), 100 mg ml$^{-1}$ primocin (InvivoGen), 0.1 mM β-mercaptoethanol (Invitrogen) and 10 ng ml$^{-1}$ of fibroblast growth factor 2 (FGF2, Invitrogen) at 5% $CO_2$ at 37 °C. The medium was refreshed daily. At ~70–80% confluency, H9 cells were differentiated into cortical organoids. After dissociation, 9,000 cells per well were plated into low-attachment V-bottom 96-well plates (Sumitomo Bakelite, MS9096V) to form aggregates in medium consisting of Glasgow's Minimal Essential Medium (GMEM, Invitrogen),

20% KSR, 0.1 nM non-essential amino acids, 100 mg ml⁻¹ primocin, 0.1 mM β-mercaptoethanol, sodium pyruvate (Invitrogen), Wnt inhibitor IWR-1-endo (Calbiochem) and TGF-β inhibitor SB431542 (Stemgent). ROCK inhibitor Y-27632 (20 μM, BioPioneer) was added in the medium from D0 to D6 to prevent cell death. From D0 to D18, the organoids were maintained at 5% $CO_2$, 37 °C, and half of the medium was changed every 2–3 d. From D18 to D35, the organoids were transferred to Petri dishes and cultured in the medium consisting of DMEM/F12 with N2 (Invitrogen), GlutaMAX, chemically defined lipid concentrate (CDLC, Invitrogen) and 0.4% methylcellulose (Sigma) at 5% $CO_2$, 40% $O_2$ and 37 °C. The medium was refreshed every 2–3 d.

**Sample preparation and scRNA-seq.** Organoids were collected at D18 (160 organoids) and D35 (25 organoids), dissociated into single cells and subjected to Evercode Cell Fixation (Parse Biosciences). The organoids were dissociated into a single-cell suspension using Papain Dissociation System (Worthington), following the manufacturer's manual. The dead cells in the single-cell suspension were removed using EasySep Dead Cell Removal (Annexin V) Kit (STEMCELL Technologies), following the manufacturer's manual. The cell suspension was then passed through a 40 mm cell strainer before assessing cell number and viability. Samples with total cell numbers >1,000,000 and >80% viability were further processed for cell fixation and freezing following Parse Biosciences User Manual. The samples were then sent to Genomics Research and Technology Hub, UCI, for barcoding and library preparation using Evercode WT kit (Parse Biosciences). Ten thousand cells per sample and 50,000 reads per cell were targeted for sequencing. The sequencing was done using NovaSeq 6000 (Illumina). Alignment was performed using Split-pipe (Parse Biosciences).

**Growth factor exposure and RT–qPCR.** Between D15 and D21, the organoids were exposed to 400 ng ml⁻¹ FGF8b or 50 ng ml⁻¹ BMP4 (with 3 mM CHIR99021) in the culture medium. Untreated organoids were used as a control group. The organoid samples were collected at D35 and lysed using Buffer RLT (Qiagen). RNA was extracted using the RNeasy Mini Kit (Qiagen), following the manufacturer's manual. Then, 1,000–3,000 ng RNA from each sample was converted to complementary DNA using SuperScript IV First-Strand Synthesis Reaction (Invitrogen). PowerTrack SYBR Green Master Mix (Applied Biosystems), cDNA and primers were mixed and loaded into 384-well plates (Invitrogen). The RT–qPCR was carried out by using QuantStudio 7 Real-Time PCR System (Applied Biosystems). The following primers were used: EOMES (amplicon size, 225 bp) forward 5′-CGACAATAACATGCAGGGCAA-3′, reverse 5′-TCATTCAAGTCCTCCACGCC-3′; PAX6 (amplicon size 48 bp) forward 5′-TGTCCAACGGATGTGTGAGTA-3′, reverse 5′-CAGTCTCGTAATACCTGCCCA-3′; CoupTF1 (NR2F1) (amplicon size 104 bp) forward 5′-ATCGTGCTGTTCACGTCAGAC-3′, reverse 5′-TGGCTCCTCACGTACTCCTC-3′; GAPDH (amplicon size 69 bp) forward 5′-CTCTCTGCTCCTCCTGTTCGAC-3′, reverse 5′-TGAGCGATGTGGCTCGGCT-3′.

**Reporting summary**
Further information on research design is available in the Nature Portfolio Reporting Summary linked to this article.

## Data availability
The human cortical organoid scRNA-seq are available at NCBI GEO at accession number GSE239542. The human pancreatic islet scRNA-seq data were originally published by Burkhardt et al.[36]; the raw gene expression counts and treatment condition metadata were downloaded from NCBI GEO at accession GSE161465. The scRNA-seq data of human COVID-19 BALF samples were originally published in Liao et al.[38]; the gene expression matrices and cell-type annotation metadata were downloaded from NCBI GEO at GSE145926. The spatial Stereo-seq of mouse embryogenesis at time E9.5 was published originally in Chen et al.[39]; the annotated spatial data were extracted from the file 'Mouse_embryo_all_stage.h5ad' hosted at https://db.cngb.org/stomics/mosta/download/.

## Code availability
FlowSig is available to install as a Python package from GitHub at https://github.com/axelalmet/flowsig. All scripts used to generate the analysis in this manuscript are available at GitHub at https://github.com/axelalmet/FlowSigAnalysis_2023. The processed versions of all datasets used in this study, including cell-type annotation and cell–cell communication output from CellChat and COMMOT for non-spatial and spatial data, respectively, are available at: https://doi.org/10.5281/zenodo.10850397 (ref. 69).

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

## Acknowledgements
This work was (partially) supported by NSF grants DMS1763272, CBET2134916 and MCB2028424, NIH grants R01AR079150 and R01DE030565, the Chan Zuckerberg Initiative grant AN-0000000062, and a grant from the Simons Foundation (594598). This work was supported by the NIH R00HD096105, NSF RECODE2225624, New Investigator Faculty Award and start-up funds from the UCI School of Medicine (M.W.), and the FRAXA Postdoctoral Fellowship (Y.C.T.). We thank C. Squires for useful discussions about the UT-IGSP algorithm and X. Wang for initial preprocessing of the cortical organoid scRNA-seq data.

## Author contributions
A.A.A. and Q.N. conceived the method. A.A.A. implemented the method. A.A.A. generated the numerical results. Y.-C.T. and M.W. generated the experimental results. A.A.A., Y.-C.T., M.W. and Q.N. interpreted the results, generated the figures and wrote the paper. All authors reviewed the manuscript.

## Competing interests
The authors declare no competing interests.

## Additional information

**Correspondence and requests for materials** should be addressed to Qing Nie.

# Reporting Summary

## Statistics

For all statistical analyses, confirm that the following items are present in the figure legend, table legend, main text, or Methods section.

| n/a | Confirmed | |
|---|---|---|
| ☐ | ☒ | The exact sample size (*n*) for each experimental group/condition, given as a discrete number and unit of measurement |
| ☒ | ☐ | A statement on whether measurements were taken from distinct samples or whether the same sample was measured repeatedly |
| ☐ | ☒ | The statistical test(s) used AND whether they are one- or two-sided<br>*Only common tests should be described solely by name; describe more complex techniques in the Methods section.* |
| ☒ | ☐ | A description of all covariates tested |
| ☒ | ☐ | A description of any assumptions or corrections, such as tests of normality and adjustment for multiple comparisons |
| ☐ | ☒ | A full description of the statistical parameters including central tendency (e.g. means) or other basic estimates (e.g. regression coefficient) AND variation (e.g. standard deviation) or associated estimates of uncertainty (e.g. confidence intervals) |
| ☐ | ☒ | For null hypothesis testing, the test statistic (e.g. *F*, *t*, *r*) with confidence intervals, effect sizes, degrees of freedom and *P* value noted<br>*Give P values as exact values whenever suitable.* |
| ☒ | ☐ | For Bayesian analysis, information on the choice of priors and Markov chain Monte Carlo settings |
| ☒ | ☐ | For hierarchical and complex designs, identification of the appropriate level for tests and full reporting of outcomes |
| ☐ | ☒ | Estimates of effect sizes (e.g. Cohen's *d*, Pearson's *r*), indicating how they were calculated |

*Our web collection on statistics for biologists contains articles on many of the points above.*

## Software and code

Policy information about availability of computer code

| Data collection | No software was used. |
|---|---|
| Data analysis | The code for the FlowSig is available on GitHub as a Python package (https://github.com/aalmet/FlowSig), along with all corresponding Python and R analysis code to reproduce the results (https://github.com/aalmet/FlowSigAnalysis_2023). The processed versions of the analyzed datasets are available at the following Zenodo repository: https://zenodo.org/doi/10.5281/zenodo.10850397<br><br>The following programming languages and versions were used:<br>Python 3.8.6<br>R 4.2.3<br><br>The following software packages and versions were used:<br>AnnData 0.9.2<br>causaldag 0.1a163<br>CellChat 1.6.1<br>CellPhoneDB 5.0.0<br>COMMOT 0.0.3<br>DIALOGUE 1.0<br>conditional_independence 0.1a6<br>graphical-model-learning 0.1a8<br>graphical-models 0.1a19<br>GraphPad Prism 9 |

joblib 1.3.1
liana 1.0.4
Matplotlib 3.7.2
Mofapy2 0.7.0
Mofax 0.3.6
multinichenetr 1.0.3
Networkx 3.1
NSF 0.0.1
Numpy 1.24.4
Omnipath 1.0.8
Pandas 2.0.3
pygam 0.8.0
pyLIGER 0.2.0
py-pde 0.36.0
PyTorch 2.0.1
Scanpy 1.9.3
Scipy 1.10.1
scITD 1.0.4
Seaborn 0.12.2
Squidpy 1.2.3

For manuscripts utilizing custom algorithms or software that are central to the research but not yet described in published literature, software must be made available to editors and reviewers. We strongly encourage code deposition in a community repository (e.g. GitHub). See the Nature Portfolio guidelines for submitting code & software for further information.

## Data

Policy information about availability of data

All manuscripts must include a data availability statement. This statement should provide the following information, where applicable:
- Accession codes, unique identifiers, or web links for publicly available datasets
- A description of any restrictions on data availability
- For clinical datasets or third party data, please ensure that the statement adheres to our policy

The human cortical organoid scRNA-seq is available at NCBI GEO at accession number GSE239542 and will be released upon publication. Gene expression count matrices were constructed by aligning FASTQ files using Split-pipe v0.9.6 by Parse Biosciences. The alignment was done using the reference genome GRCh38. The human pancreatic islet scRNA-seq data was originally published by Burkhardt et al.38; the raw gene expression counts and treatment condition metadata were downloaded from NCBI GEO at accession GSE161465. The scRNA-seq data of human COVID-19 patient BALF samples was originally published in Liao et al.40; the gene expression matrices and cell type annotation metadata were downloaded from NCBI GEO at GSE145926. The spatial Stereo-seq of mouse embryogenesis at time E9.5 was published originally in Chen et al.41; the annotated spatial data was extracted from the file "Mouse_embryo_all_stage.h5ad" hosted at https://db.cngb.org/stomics/mosta/download/.

## Human research participants

Policy information about studies involving human research participants and Sex and Gender in Research.

| Reporting on sex and gender | We used public data in the manuscript and human embryonic stem cell lines and there is no human research involved. |
| Population characteristics | We used public data in the manuscript and human embryonic stem cell lines and there is no human research involved. |
| Recruitment | We used public data in the manuscript and human embryonic stem cell lines and there is no human research involved. |
| Ethics oversight | We used public data in the manuscript and human embryonic stem cell lines and there is no human research involved. |

Note that full information on the approval of the study protocol must also be provided in the manuscript.

# Field-specific reporting

Please select the one below that is the best fit for your research. If you are not sure, read the appropriate sections before making your selection.

☒ Life sciences  ☐ Behavioural & social sciences  ☐ Ecological, evolutionary & environmental sciences

For a reference copy of the document with all sections, see nature.com/documents/nr-reporting-summary-flat.pdf

# Life sciences study design

All studies must disclose on these points even when the disclosure is negative.

| Sample size | No statistical sample size calculation was performed for this study. Instead, the sample sizes were determined based on practical |

| Sample size | considerations and the need to obtain a sufficient number of cells for subsequent analyses. For sequencing: at day 18 and day 35, two sets of samples were collected. We pooled 160 cortical organoids into one sample for D18, and we pooled 25 cortical organoids into one sample for D35. These numbers was chosen to ensure that we could obtain more than 1 million cells per sample, which was necessary to achieve sufficient material for the fixing and freezing steps prior to sequencing. For RT-qPCR: samples were pool from 4 organoids per sample and 2-4 samples per group. The rationale for these sample sizes is that RT-qPCR is a highly sensitive assay capable of detecting changes in gene expression even with relatively small sample sizes. Our results confirmed that pooling organoids in this manner provided enough RNA to detect meaningful differences in gene expression. |
|---|---|
| Data exclusions | Cells expressing fewer than 500 unique genes, or more than 10,000 genes were removed. Cells with more than 5% of their total gene expression contributed by mitochondrial genes were removed. No data exclusion for RT-qPCR experiment. |
| Replication | All experiments were at least in duplicates. All attempts at replication were successful and included in data analyses. |
| Randomization | Randomization is not relevant to this study. For sequencing: the samples had to be collected at specific time points and processed into single cell suspensions immediately. After all single cell suspensions were gathered, these samples were sent to sequencing core all together. For RT-qPCR: the control and experimental groups were from the organoids produced in the same batch. Sample preparations for control and experimental groups were collected at day 35 and RNA extractions were performed together. |
| Blinding | Blinding is not necessary for this study. For sequencing: the samples had to be collected at specific time points and processed into single cell suspensions immediately. Afterwards, the samples were handled by the core facility staff and the sequencing data handled by researchers who had no prior assumption of the biology, as this is an exploratory study. For RT-qPCR: RNA extractions were done for all samples at the same time in no particular order. Blinding is not relevant to the execution of RT-qPCR and technical replicates were included to avoid bias. Data analysis was done in the same way for all groups with no data exclusion. |

# Reporting for specific materials, systems and methods

We require information from authors about some types of materials, experimental systems and methods used in many studies. Here, indicate whether each material, system or method listed is relevant to your study. If you are not sure if a list item applies to your research, read the appropriate section before selecting a response.

## Materials & experimental systems

| n/a | Involved in the study |
|---|---|
| ☒ | ☐ Antibodies |
| ☐ | ☒ Eukaryotic cell lines |
| ☒ | ☐ Palaeontology and archaeology |
| ☒ | ☐ Animals and other organisms |
| ☒ | ☐ Clinical data |
| ☒ | ☐ Dual use research of concern |

## Methods

| n/a | Involved in the study |
|---|---|
| ☒ | ☐ ChIP-seq |
| ☒ | ☐ Flow cytometry |
| ☒ | ☐ MRI-based neuroimaging |

## Eukaryotic cell lines

Policy information about cell lines and Sex and Gender in Research

| Cell line source(s) | Cortical organoids were generated using H9 (WA09; WAe009-A; WiCell) human embryonic stem cell. H9 originated from a female. |
|---|---|
| Authentication | The cell line was purchased directly from the WiCell under the MTA, has been tested negative for viral and mycoplasma infections, and regularly checked for chromosomal abnormalities. The H9 is de-identified for distribution from the WiCell. |
| Mycoplasma contamination | The H9 cell line was tested negative for mycoplama contamination. |
| Commonly misidentified lines (See ICLAC register) | No commonly misidentified lines. |

