## [Peer Review File · Nature Methods]

Peer Review Information

Manuscript Title: Inferring pattern-driving intercellular flows from single-cell and spatial transcriptomics

Corresponding author name(s): Qing Nie

Editorial Notes:

Reviewer Comments & Decisions:

Decision Letter, initial version:
--

16th Dec 2023

Dear Prof Nie,

We appreciate your patience when your Article, "Inferring pattern-driving intercellular flows from single-cell and spatial transcriptomics" is sent out for peer review. This paper has now been seen by 2 reviewers. As you will see from their comments below, although the reviewers find your work of potential interest, they have raised a number of concerns. We are interested in the possibility of publishing your paper in Nature Methods, but would like to consider your response to these concerns before we reach a final decision on publication.

We therefore invite you to revise your manuscript to address all the concerns raised by our reviewers.

- * include a point-by-point response to the reviewers and to any editorial suggestions
- * please underline/highlight any additions to the text or areas with other significant changes to

facilitate review of the revised manuscript

- * address the points listed described below to conform to our open science requirements
- * ensure it complies with our general format requirements as set out in our guide to authors at www.nature.com/naturemethods
- * resubmit all the necessary files electronically by using the link below to access your home page

[REDACTED]

We hope to receive your revised paper within 4 months. If you cannot send it within this time, please let us know. In this event, we will still be happy to reconsider your paper at a later date so long as nothing similar has been accepted for publication at Nature Methods or published elsewhere.

OPEN SCIENCE REQUIREMENTS

REPORTING SUMMARY AND EDITORIAL POLICY CHECKLISTS

DATA AVAILABILITY

We strongly encourage you to deposit all new data associated with the paper in a persistent repository where they can be freely and enduringly accessed. We recommend submitting the data to discipline-

specific and community-recognized repositories; a list of repositories is provided here:
<http://www.nature.com/sdata/policies/repositories>

All novel DNA and RNA sequencing data, protein sequences, genetic polymorphisms, linked genotype and phenotype data, gene expression data, macromolecular structures, and proteomics data must be deposited in a publicly accessible database, and accession codes and associated hyperlinks must be provided in the “Data Availability” section.

Please include a “Data availability” subsection in the Online Methods. This section should inform readers about the availability of the data used to support the conclusions of your study, including accession codes to public repositories, references to source data that may be published alongside the paper, unique identifiers such as URLs to data repository entries, or data set DOIs, and any other statement about data availability. At a minimum, you should include the following statement: “The data that support the findings of this study are available from the corresponding author upon request”, describing which data is available upon request and mentioning any restrictions on availability. If DOIs are provided, please include these in the Reference list (authors, title, publisher (repository name), identifier, year). For more guidance on how to write this section please see:
<http://www.nature.com/authors/policies/data/data-availability-statements-data-citations.pdf>

CODE AVAILABILITY

Please include a “Code Availability” subsection in the Online Methods which details how your custom code is made available. Only in rare cases (where code is not central to the main conclusions of the paper) is the statement “available upon request” allowed (and reasons should be specified).

For more information on our code sharing policy and requirements, please see:
<https://www.nature.com/nature-research/editorial-policies/reporting-standards#availability-of-computer-code>

MATERIALS AVAILABILITY

ORCID

Sincerely,

Lin Tang, PhD
Senior Editor
Nature Methods

Reviewers' Comments:

Reviewer #1:

Remarks to the Author:

Review for the manuscript: 'Inferring pattern-driving intercellular flows from single-cell and spatial transcriptomics'.

Almet et al. introduce FlowSig, a computational method that aims to capture information flow in multicellular networks (based on either scRNA-seq or ST data), by integrating information about cell-cell communication and intracellular gene expression within a graphical causal modeling framework. They suggest to integrate two often complementary computational views of these data - gene expression modules (GEMs) and ligand-receptor interactions (LR) networks. Specifically, given GEMs and LR networks as input, they aim to identify intercellular flow driven by LR interactions by learning a corresponding CPDAG, and use additional biological prior information and constraints (e.g. downstream TF targets and perturbation data for scRNA-seq data, spatial location of relative signaling for ST data), to reduce false discovery.

The authors show good performance of FlowSig on synthetic data and newly generated cortical organic data, as well as on stimulated scRNA-seq data of pancreatic islets, COVID-19 data corresponding to varying severity of the disease, and mouse embryogenesis.

The approach tackles an important and understudied problem related to inferring overall information flow patterns in single-cell and spatial data. The paper is well written although there are some over-selling aspects and key details about the method are missing from the text (mentioned below). It would be useful to clarify and elaborate on the method in the text. The manuscript would also benefit from adding baselines and benchmarking results.

Comments:

- It's not clear from the description of the methods how inflows and outflows from different cells are integrated.

- In the abstract it says 'We benchmark FlowSig..' but there are no comparisons to alternative methods in the manuscript (only comparisons similar to ablation studies, where a subset of the method is used). It would be useful to add such benchmarks.

- For the section on synthetic validation of FlowSig, it's not clear how restricted is the hypothesis class, meaning, what are the complete set of nodes in each of the three scenarios over which edges are learned.

- It's not clear why the statement "we can use the scRNA-seq data to validate FlowSig and treat the D35 data as a "perturbation" of the "control" D18 data" is justified. Why can it be treated as a perturbation? It would be useful to be more exact here, on what underlying assumptions are you relying or will need for the analysis of these two time points.

- Elaborate on the distinction between inflowing and outflowing signals (how are they differentiated) in : "Differential expression analysis identified 26 differentially expressed inflowing (received) signals (Fig. 3a), and 16 differentially expressed outflowing (sent) signals"

- Line 170 - again unclear how to reason about differentially inflowing/outflowing signals - at a minimum refer to methods.

- "We used these cell type annotations as input for preliminary CellChat analysis." - how?

- Lines 220-224 - unclear how the two networks are constructed (and therefore, the follow up reasoning and interpretation in the text is unclear).

- "extracted the top ten TFs by module membership" - how?

- "where we ranked upstream TFs based on feature (Gini) importance relative to all potential upstream TFs of Shh" - how?

- "This module resembles the well-known "activator-inhibitor" system that can generate potential Turing patterns" - elaborate? In general it would be useful to tie this analysis/discussion closer to patterning.

- This is also related to the first sentence in the discussion. You write "We developed a method, FlowSig, to infer and describe how intercellular communication drives tissue-scale gene expression patterning", but it seems like it's overselling the method in a way, because currently there's no direct link to patterning. The output of the method is interesting and useful enough as it is, no need to overstate this.

- It would be useful to see an extended discussion on limitations (beyond the static related discussion)
 - where do you expect your method to fail or struggle?

- Also, it would be very useful to include a short example, or steps, in the methods of how to go from input (e.g. ST or scRNA-seq data) to a network structure (in a detailed way). It is currently difficult to understand the actual steps (at the level of reproducing results) from the text description.

Minor Comments:

- In Fig 1, the symbols/letters in the output boxes are not defined

- Figure references in the section about pancreatic islets refer to fig. 3 instead of 4 (and this propagates to the following figures, e.g. in the covid section the ref is to fig. 4).

- Lines 256-259 - which inflowing signals (and GEMs) are removed? are they associated with biological processes that are related to covid progression?

- Fig 5 d-f are visually hard to interpret, g-h: explain what the horizontal bars at the bottom represent.

Typos (or, better to rephrase):

- means that are relatively
- we aggregated inflowing signals were ranked
- from the different perturbation data
- it will be to adapt
- respectively. and GEM construction
- that are based the concepts
- that have same skeleton
- we then take calculate the

Reviewer #2:

Remarks to the Author:

Summary of the key results:

The authors present a computational method that extracts cell communication patterns ("information flow") across three cellular entities, extending this paradigm from the pair of cell types that is often studied. This idea is relevant and there is novelty in its execution. Some context in the related literature is missing as noted below and would add value to this manuscript. Validation was attempted in line with what is possible in cell communication modeling. Baseline models are largely missing but would add contextualization in the literature.

Comments:

- Contextualization in existing work: The proposed approach centered on in+out flows may be partially reflected in multicellular representation learning, in particular DIALOGUE [reference 8 in the manuscript] and scITD (<https://www.biorxiv.org/content/10.1101/2022.02.16.480703v1>). Here, an in-out flow could be a pattern across three cell types for example. It would both be worth discussing how this communication path of length 3 is reflected in multicellular programs. As a note for the discussion aspect, I find that the value added to the field by DIALOGUE is somewhat underrepresented by "to construct gene expression modules (GEMs) defined by gene sets such that intra-set gene expression is more correlated than genes between sets."
- The proposed model depends on GEMs and signal to be predetermined, which is done based on other models, eg pyLIGER and CellChat in the organoid example. This pipeline presents risks in terms of error propagation and ambiguity in terms of choice of tool for each of these two pre-processing steps. An extended discussion of robustness would be appreciated.
- On baseline models: While appreciating aspects of the novelty of this approach, I am missing baseline models. First, DIALOGUE and scITD could provide baseline for some aspects here, eg correctly grouping in and out flows into a multicellular program (without directionalily). Second, basic analyses, such as connecting CellChat signals to in and out flow if they are connected through the prior could add additionally perspective on the added value of the model proposed here. I could imagine that even more baselines are in reach.
- Continuing on baselines: The quality of insight "FlowSig identified that FGF, IL6, MDK, and SST were the dominant drivers of intercellular flows " could also be derived from a cell communication model of pairs of cell types like CellChat, so also here, demonstrating how this insight is reproduced / improved would make this more convincing.
- Similarly, it would be interesting to discuss further how this method adds value over CellChat or other cell communication models in the COVID example, ie in the case where communication patterns are not stimulated, and thus, harder to interpret than in the previous "validation" examples in the manuscript. For example, the result "In this case, FlowSig found that increasing severity of COVID-19 is associated with 1) a gradual loss of regulatory intercellular inflows and 2) an increase of inflammatory chemokine outflow that is driven by macrophages and neutrophils." could be contrasted with a baseline. One potential issue that I see is that the flow modules carry more information than pairwise associations (good!), but may also be more complicated to interpret. This could be addressed by showcasing how this new method yield 1) new results and 2) usable results among the new results, specifically in this example.
- Glancing over the Github repo, the code seems well maintained.

Clarity and context: lucidity of abstract/summary, appropriateness of abstract, introduction and conclusions:

- I was not able to access the code under the described URL "<https://github.com/axelalmet/cellflowsig>" but found "<https://github.com/axelalmet/flowsig>" which I assume is this work.

Author Rebuttal to Initial comments**Response to Reviewer #1**

Reviewer #1:

Remarks to the Author:

Review for the manuscript: 'Inferring pattern-driving intercellular flows from single-cell and spatial transcriptomics'.

Almet et al. introduce FlowSig, a computational method that aims to capture information flow in multicellular networks (based on either scRNA-seq or ST data), by integrating information about cell-cell communication and intracellular gene expression within a graphical causal modeling framework. They suggest to integrate two often complementary computational views of these data - gene expression modules (GEMs) and ligand-receptor interactions (LR) networks. Specifically, given GEMs and LR networks as input, they aim to identify intercellular flow driven by LR interactions by learning a corresponding CPDAG, and use additional biological prior information and constraints (e.g. downstream TF targets and perturbation data for scRNA-seq data, spatial location of relative signaling for ST data), to reduce false discovery.

The authors show good performance of FlowSig on synthetic data and newly generated cortical organic data, as well as on stimulated scRNA-seq data of pancreatic islets, COVID-19 data corresponding to varying severity of the disease, and mouse embryogenesis.

The approach tackles an important and understudied problem related to inferring overall information flow patterns in single-cell and spatial data. The paper is well written although there are some over-selling aspects and key details about the method are missing from the text (mentioned below). It would be useful to clarify and elaborate on the method in the text. The manuscript would also benefit from adding baselines and benchmarking results.

Comments:

- It's not clear from the description of the methods how inflows and outflows from different cells are integrated.

We apologize for the confusion. In the revised Methods section, we have expanded our description explaining how inflows and outflows from different cells are integrated.

Specifically, on page 30, we have further clarified how each cell is associated a vector constructed from three types of measurements: signal inflow measurements, which are receptor gene expression weighted by the average expression of their known downstream transcription factor genes, intracellular "module" enrichment, which is the cell's membership weight to a multigene set module (inferred from, e.g. matrix factorization), which measures how strongly the cell expresses those genes in the module, and signal outflow, which is simply defined as ligand gene expression.

We clarify further that when measuring signal inflow, we are not measuring *from which cells* the signals were sent, but rather, how much signal *has been received by the cell*. Similarly, when measuring signal outflow, we are not measuring how much of the expressed signal ligand was actually received by other cells (as measured by, e.g., signal inflow), but simply how much of the signal the cell is expressing.

- In the abstract it says ‘We benchmark FlowSig..’ but there are no comparisons to alternative methods in the manuscript (only comparisons similar to ablation studies, where a subset of the method is used). It would be useful to add such benchmarks.

Thank you for this suggestion. In our revised manuscript, on pages 8–11, we included comparisons to methods with partial overlap to FlowSig, including DIALOGUE, MOFACellular, MOFAtalk, MultiNicheNet, scITD, and Tensor-cell2cell. These methods construct “multicellular representations”, some of which can be interpreted as intercellular flows. We include a summary of our comparison in a new section of the revised Results section, “Benchmarking FlowSig against other multicellular representation methods” and a complete description of how we implemented each method and the method’s results in the Supplementary Information.

- For the section on synthetic validation of FlowSig, it’s not clear how restricted is the hypothesis class, meaning, what are the complete set of nodes in each of the three scenarios over which edges are learned.

Thank you for this comment. In the revised Results, on page 6, we clarified the sets of nodes over which we were learning directed relations.

In the first simulation model that describes SHH-dependent BMP4 outflow, FlowSig learnt intercellular flows for a network containing five nodes: diffusible SHH ligand, unbound PTCH1 receptor, inflowing SHH due to SHH-PTCH1 binding, FOXF1 TF, and diffusible BMP4 ligand. In the second simulation model, which describes SHH-driven patterning via transcription factor expression, FlowSig learnt intercellular flows over a network containing seven nodes: diffusible SHH ligand, unbound PTCH1 receptor, inflowing SHH (SHH-PTCH1 bound complex), and the four TFs NKX2.2, OLIG2, PAX6, and IRX3. For the third model, which simulates competing inflows from SHH and BMP4 to drive dorsoventral patterning, intercellular flows were learnt over a set of nine nodes, including diffusible SHH ligand, unbound PTCH1 receptor, inflowing SHH (bound SHH-PTCH1 complex), diffusible BMP4 ligand, unbound BMP1A+BMPR2 receptor, inflowing BMP4 (bound BMP complex), and three TF variables, dorsal D, intermediate I, and ventral V

- It's not clear why the statement "we can use the scRNA-seq data to validate FlowSig and treat the D35 data as a "perturbation" of the "control" D18 data" is justified. Why can it be treated as a perturbation? It would be useful to be more exact here, on what underlying assumptions are you relying or will need for the analysis of these two time points.

Thank you for this comment. In the revised manuscript, on page 12, we have explained why we chose to consider D35 as a "perturbation".

At D18, the organoids self-generate opposing spatial profiles of FGF and BMP signaling. It is these FGF and BMP signaling patterns that drive drastic changes in both cell type composition and gene transcription from D18 to D35 as the organoid populations mature and differentiate. Therefore, rather than treat both D18 and D35 timepoints as samples of the same "steady state cell type population", we decided to treat the D35 timepoint as a "perturbation" of the D18 timepoint with respect to both cell type, transcriptional, and thus signaling composition.

- Elaborate on the distinction between inflowing and outflowing signals (how are they differentiated) in : "Differential expression analysis identified 26 differentially expressed inflowing (received) signals (Fig. 3a), and 16 differentially expressed outflowing (sent) signals"

Thank you for this suggestion. In both the revised Methods on page 34, we provided more detail about our differential expression analysis of signal inflow and outflow.

We performed differential expression analysis of signal inflow variables (R^*TF) between conditions, and differential expression analysis of signal outflow variables (L) between conditions, calculated using a nonparametric Mann-Whitney U test. In the former, we only considered signal inflow variables. In the latter, we only considered signal outflow variables for analysis. Then, we retained variables for which their log-fold change was above a specified threshold (e.g., $\log_2 FC \geq 0.5$) meaning that the variable had significantly been up or downregulated due to perturbation, and if their adjusted p-value after correction for multiple hypothesis testing was below a specified threshold (e.g. $p - adj. < 0.05$), measuring statistical significance in some way.

- Line 170 – again unclear how to reason about differentially inflowing/outflowing signals – at a minimum refer to methods.

Thank you for this suggestion. Please see our previous response.

- "We used these cell type annotations as input for preliminary CellChat analysis." – how?

Thank you for this question. We have clarified in the revised Methods section on page 32 and in the revised Results on page 15 how we used cell type annotation to infer the initial preliminary list of ligand-receptor interactions from CellChat.

CellChat infers significant ligand-receptor interactions between prespecified groups defined by cell type annotations. CellChat calculates the interaction score using a mass-action-based formula for every significantly expressed ligand-receptor interaction pair. We then use the list of ligand-receptor interactions to construct the initial candidate sets of signal inflow and signal outflow variables.

- Lines 220-224 – unclear how the two networks are constructed (and therefore, the follow up reasoning and interpretation in the text is unclear).

We apologize for the lack of clarity. In the revised manuscript (Results and Methods), we explained exactly how these two condition-specific intercellular flow networks were constructed on pages 16–17.

To construct the two condition-specific networks, one describing the intercellular flows in the control Ctrl condition and one describing the intercellular flows in the stimulated IFN- γ conditions, we first extracted the signal outflow variables that were differentially expressed, i.e., upregulated, for that condition. We then backtrack through the global intercellular flow network inferred by FlowSig to extract the gene expression modules that are connected to the differentially expressed signal outflow variables and then the signal inflow variables that are connected to these gene expression module nodes. This procedure results in two subnetworks that describe how intercellular flows drive the upregulated signal outflows for each condition.

- “extracted the top ten TFs by module membership” – how?

Thank you for this comment. In our revised manuscript, we clarified in the revised Methods on pages 35–36.

We use the fact that, for each biological condition, pyLIGER uses matrix factorization to decompose the $N^{(c)}$ -by- G gene expression matrix, $X^{(c)}$, into K gene expression modules via $X^{(c)} = F^{(c)} * (W + V^{(c)})^T$, where $F^{(c)}$ is a $N^{(c)}$ -by- K matrix describing the condition-specific cell weights to the K modules, W is a G -by- K matrix describing the condition-shared gene weights to the module, while $V^{(c)}$ is a G -by- K matrix describing the condition-specific gene weights to the matrix. That is, higher values of $W + V^c$ indicate stronger contribution to the gene expression modules. Therefore, for each module, k , we rank genes and thus transcription factors by their

total weight $W_{:,k} + V_{:,k}^{(c)}$. Similarly, the spatial factorization also decomposes the gene expression matrix into $X = F * W^T$, where we now rank genes and TFs in increasing order of their weight $W_{:,k}$ for each spatial module k .

- “where we ranked upstream TFs based on feature (Gini) importance relative to all potential upstream TFs of *Shh*” – how?

Thank you for this question. We revised the manuscript to include a more detailed description in the Methods section on page 37 and referenced the relevant section in the Results on page 22.

For each signal outflow variable, we use the inferred FlowSig network to extract the spatial gene expression modules that connect to this variable. For each module connecting to the signal outflow variable, we extract the top contributing transcription factors by module weight (see previous response). The union of these transcription factor genes forms the candidate set of potential upstream regulators of the signal. We then adapt the approach by Cang *et al.* (published in Nature Methods 2023) and use a random forest regression model to model signal outflow, i.e. signal ligand gene expression, as a function of upstream transcription factor gene expression. We then use the model to rank each potential upstream transcription factor by their feature importance, which, for a random forest model, is calculated by the Gini importance (also known as the mean decrease in impurity).

- “This module resembles the well-known “activator-inhibitor” system that can generate potential Turing patterns” - elaborate? In general it would be useful to tie this analysis/discussion closer to patterning.

Thank you for this suggestion. In the revised Results section on pages 22–23, we have clarified more on this analogy.

Activator-Inhibitor systems describe how interactions between signals (often called “morphogens”) can generate complex patterning via cell fate. They have three key features. First, at least one of the signals propagates over space—in our analysis of the E9.5 Stereo-seq data, both *Shh* and *Wnt5a* signal ligands propagate via diffusion. Second, one of the signals—in this case, *Shh*—upregulates both itself and the other signal, *Wnt5a*. Third, the other signal, *Wnt5a*, downregulates the “activating” signal, *Shh*. Using additional downstream analysis, we inferred that *Shh* promotes itself via the transcription factor, *Foxa2*, and activates *Wnt5a* via the transcription factors, *Foxa2*, *Nkx6-1*, and *Sox21*, while *Wnt5a* inhibits *Shh* via the transcription factor *Myc*. Therefore, in the E9.5 dataset, we identified all potential ingredients for a two-signal

activator-inhibitor system. In the modeling literature, it is well known that such systems are able to generate “Turing patterns”, that are defined by their complex spatial variation. These Turing pattern systems have been suggested to drive cell fate patterning in development, in which both *Shh* and *Wnt5a* play key roles.

- This is also related to the first sentence in the discussion. You write “We developed a method, FlowSig, to infer and describe how intercellular communication drives tissue-scale gene expression patterning”, but it seems like it’s overselling the method in a way, because currently there’s no direct link to patterning. The output of the method is interesting and useful enough as it is, no need to over-state this.

Thank you for the suggestion. Based on the suggestion, we have toned down our description in the revised Discussion section

- It would be useful to see an extended discussion on limitations (beyond the static related discussion) - where do you expect your method to fail or struggle?

Thank you for the useful feedback. We have included an extended discussion of FlowSig in the revised Discussion section on pages 27–28.

There are several cases, where we may expect FlowSig to struggle or fail. First, when one is trying to infer intercellular flows across an extremely high number of variables, i.e., on the order of several hundreds or thousands, or millions, as it is known that for causal structure learning methods, including FlowSig’s underlying method, UT-IGSP, the number of false positive inferred edges increases as the number of variables increases. Second, the sample size is not sufficiently large, FlowSig will not be able to determine the undirected dependent edges accurately from conditional independence and conditional invariance testing. Third, when analyzing non-spatial scRNA-seq, if there are any perturbations that completely knock out a signal, i.e., set the expression of a signal outflow (or inflow) to be uniformly zero, FlowSig cannot learn the resulting intercellular flow as the standard deviation of the variable will be zero, causing the partial correlation testing step to fail. On the other hand, if the perturbation is not sufficiently “strong enough” to induce a change in distribution, conditional invariance testing may fail. Fourth, as FlowSig uses partial correlation for conditional independence testing, and thus assumes that the data can be modeled using a linear Gaussian model, any underlying intercellular flows generating data that significantly violate this assumption may not be inferred by FlowSig.

- Also, it would be very useful to include a short example, or steps, in the methods of how to go from input (e.g. ST or scRNA-seq data) to a network structure (in a detailed way). It is currently difficult to understand the actual steps (at the level of reproducing results) from the text

description.

Thank you for the suggestion. In the revised Methods, on pages 33–34, we have included a clearer step-by-step outline of how to go from input to network structure.

Minor Comments:

- In Fig 1, the symbols/letters in the output boxes are not defined

Thank you. We have fixed this in our revised Figure 1.

- Figure references in the section about pancreatic islets refer to fig. 3 instead of 4 (and this propagates to the following figures, e.g. in the covid section the ref is to fig. 4).

Thank you for the pickup. We have fixed the Figure references in the revised Results.

- Lines 256-259 - which inflowing signals (and GEMs) are removed? are they associated with biological processes that are related to covid progression?

Thank you for the question. We have clarified which inflowing signals and GEMs are lost in the revised description of Fig. 5 on pages 19–20.

We clarify that, with respect to Healthy Controls, in Moderate COVID-19, signal inflow via AXL, CD4, F2RL1, ITGAX+ITGB2, TNFRSF12A and TNFRSF14 no longer drive intercellular flows but now inflow through CAP1 does. With respect to Moderate COVID-19, in Severe COVID-19, signal inflow via CD27, CXCR3, FPR1, IL6R+IL6ST, LTBR, NCL, NRP2+PLXNA2, SDC1, TNFRSF13B, TNFRSF17, and TNFRSF25 no longer drive intercellular flows, but like in Healthy Controls, inflow via AXL, F2RL1, and, TNFRSF14 drive intercellular flows, as does inflow into CD4.

We also note that between Healthy and Moderate COVID-19, GEM-4, GEM-10, GEM-12, GEM-14 do not mediate signal outflows, while GEM-7 does. We observe that GEM-4 is associated with Epithelial cells, GEM-10 is associated with Plasma cells and T cells, GEM-12 is associated with Macrophages and Neutrophils and Severe COVID-19, while GEM-7 is associated with Mast cells, suggesting that Moderate COVID-19 is associated with an increase in the role of Mast cells but a reduction in role of Epithelial cells and several immune cell types, including Plasma cells, T cells, Macrophages, and Neutrophils. From Moderate to Severe COVID-19, the

GEMs GEM-1, GEM-2, GEM-5, GEM-11, GEM-13, GEM-18, and GEM-19 no longer mediate signal outflows, but now GEM-12 and GEM-14 do. We observe that GEM-1, which is associated with Macrophages and mDC cells, GEM-2 is associated with B, Plasma, and pDC cells, GEM-5 is associated with Epithelial cells, GEM-11 is T cells, GEM-13 is associated with B, NK, and T cells, GEM-18 is associated with Epithelial cells, and GEM-19 is enriched for all conditions and cell types, while GEM-12 is associated with Macrophages and Neutrophils. This suggests that severity of COVID-19 is linked to the interplay between different cell types, the signals they secrete and their regulation of such signals.

- Fig 5 d-f are visually hard to interpret, g-h: explain what the horizontal bars at the bottom represent.

Thank you for this comment. In the revised manuscript, we have modified Fig. 5d–f to make the networks easier to interpret and provided an explanation of the significance of Fig. 5g–h.

Typos (or, better to rephrase):

- means that are relatively
- we aggregated inflowing signals were ranked
- from the different perturbation data
- it will be to adapt
- respectively. and GEM construction
- that are based the concepts
- that have same skeleton
- we then take calculate the

Thank you for pointing out these typos. We have fixed all of these in the revised manuscript.

Response to Reviewer #2

Remarks to the Author:

Summary of the key results:

The authors present a computational method that extracts cell communication patterns ("information flow") across three cellular entities, extending this paradigm from the pair of cell types that is often studied. This idea is relevant and there is novelty in its execution. Some context in the related literature is missing as noted below and would add value to this manuscript. Validation was attempted in line with what is possible in cell communication modeling. Baseline models are largely missing but would add contextualization in the literature.

Comments:

- Contextualization in existing work: The proposed approach centered on in+out flows may be partially reflected in multicellular representation learning, in particular DIALOGUE [reference 8 in the manuscript] and scITD (<https://www.biorxiv.org/content/10.1101/2022.02.16.480703v1>). Here, an in-out flow could be a pattern across three cell types for example. It would both be worth discussing how this communication path of length 3 is reflected in multicellular programs. As a note for the discussion aspect, I find that the value added to the field by DIALGOUE is somewhat underrepresented by "to construct gene expression modules (GEMs) defined by gene sets such that intra-set gene expression is more correlated than genes between sets."

Thank you for this comment. In the revised manuscript, on page 26, we expanded our Discussion section to provide justified and detailed explanations of what methods, such as DIALOGUE and scITD, and others, such as MOFAcellular, MOFAtalk, MultiNicheNet, and Tensor-cell2cell, extract from scRNA-seq and ST data and the insights they can reveal. We show these methods complement the insights provided by FlowSig.

- The proposed model depends on GEMs and signal to be predetermined, which is done based on other models, eg pyLIGER and CellChat in the organoid example. This pipeline presents risks in terms of error propagation and ambiguity in terms of choice of tool for each of these two pre-processing steps. An extended discussion of robustness would be appreciated.

Thank you for this suggestion. In our revised manuscript, on pages 11-12, we investigated how FlowSig's results changed when we used either a different cell-cell communication models (CellChat vs. CellPhoneDB) and a difference GEM construction method (pyLIGER vs. cNMF) for the same dataset of stimulated PBMCs as generated in Kang *et al.*

We found that due to the lack of overlap between cell-cell communication models, FlowSig's results do indeed change due to the changing sets of signal inflow and signal outflow nodes depending on the method. This stems from the lack of overlap between ligand-receptor databases used by the different cell-cell communication methods resulting in a significantly different ligand-receptor interactions inferred.

We observed that the sets of signal inflow and outflow nodes that FlowSig infers to participate in intercellular flows largely overlap and does not change significantly if one uses cNMF rather than pyLIGER for GEM construction. Between the two cases, FlowSig infers that five signal inflow variables drive intercellular flows. In the case when cNMF is used, FlowSig does not infer PLAUR inflow as a driver of intercellular flows. Similarly, of the total of 13 signal outflow variables that FlowSig inferred to be driven by intercellular flows, where nine were inferred in both cases. In this case, using FlowSig with pyLIGER identified CCL4, GZMA, and IL1B as signal outflows, while using FlowSig with cNMF identified CXCL9 as a signal outflow.

Furthermore, even though the GEMs constructed by pyLIGER and cNMF are evidently different, when we examined the intercellular flows between common signal inflow and outflow variables, we found that, for every signal outflow variable, FlowSig infers directed paths from signal inflow variables through GEMs that consist of the same transcription factors. For example, while FlowSig with pyLIGER inferred that inflow through C5AR1 drives CXCL10 outflow via GEM-1, GEM-10, GEM-11, and GEM-20, and FlowSig with cNMF inferred that inflow through C5AR1 drives CXCL10 outflow via one module, cNMF-6, these sets of GEMs shared 7 transcription factors, most notably CREM. This suggests that even with a different GEM method, FlowSig will still infer intercellular flows through GEMs that the same underlying regulatory transcription factors.

Ultimately, as different cell-cell communication inference methods and GEM inference methods are chosen primarily based on factors such as user preference and familiarity (e.g., with programming language), rather than any one of these methods being the "best" per se, we have extended FlowSig's functionality to allow for additional cell-cell communication methods, including CellPhoneDB and LIANA (which infers ligand-receptor interactions based on consensus aggregation from six methods), as well as additional GEM construction methods, such as cNMF, to be used as input.

- On baseline models: While appreciating aspects of the novelty of this approach, I am missing baseline models. First, DIALOGUE and scITD could provide baseline for some aspects here, eg correctly grouping in and out flows into a multicellular program (without directionality).

Thank you for this comment. In the revised manuscript, on pages 8–11, we have revised our Results to include a comparison of FlowSig against other baseline models.

We benchmarked FlowSig against DIALOGUE and scITD, as well as other methods, including MOFACellular, MOFATalk, MultiNicheNet, and Tensor-cell2cell, which also construct multicellular programs from scRNA-seq data. We applied all of these methods to a dataset of Interferon- β -stimulated PBMCs sampled from lupus patients that was generated by Kang *et al.* (2018). These results have been described in full in the Supplementary Materials and summarized in a new Results section, “Benchmarking FlowSig against other multicellular representation methods”.

DIALOGUE identified one MCP enriched across three cell types, CD14+ Monocytes (CD14), CD8+ T cells (CD8T), and B cells (B), suggesting the possibility of coordinated intercellular flows between these three cell types. Unfiltered CellChat output also identified 6, 886 possible inflow-outflow paths across either condition across these three cell types, suggesting that cell-cell communication may mediate the coordination between these three cell types. Both scITD and MOFACellular identified one latent factor (GEM) significantly associated with the stimulation condition and both identified that expression of the genes CXCL11, IFI6, IFIT1, IFIT3, ISG15, MX1, and TNFSF10 were significantly altered due to stimulation. Of these genes, FlowSig inferred that CXCL11 and TNFSF10 were significantly upregulated signal outflows due to stimulation. MOFATalk, MultiNicheNet, scITD and Tensor-cell2cell all identified the common interactions: CCL2 – CCR1, CCL2 – CCR5, CCL8 – CCR1, CCL8 – CCR5. Of these interactions, FlowSig inferred that inflow to CCR1 via CCL8 was a significant driver of intercellular flows across both conditions.

Second, basic analyses, such as connecting CellChat signals to in and out flow if they are connected through the prior could add additionally perspective on the added value of the model proposed here. I could imagine that even more baselines are in reach.

Thank you for this comment. Throughout the revised manuscript, we further elaborated on our discussion what FlowSig builds on compared to using CellChat alone, using the Kang *et al.* dataset as a specific motivating example on page 8.

CellChat identified 3, 445 ligand-receptor interactions across both control and stimulated conditions. Using the direct CellChat output alone, we identified 7, 209 inflow-to-outflow relationships encoded by the pair of ligand-receptor interactions, $(L_1 - R_1, L_2 - R_2)$, for which there existed a cell type triplet, (A, B, C) , such that A communicates with B via the ligand-receptor interaction $L_1 - R_1$ and B communicates with C via $L_2 - R_2$. Analysis of CellChat output implied

that there are 7, 209 inflow-to-out-flow relationships across either control or stimulated condition. Restricting output to only paracrine intercellular flows, i.e., where A , B , and C are all different cell types but the ligand-receptor interactions may be the same, CellChat output implies that there are a total of 6, 886 intercellular flows, with 3, 167 shared across both conditions, 1, 511 flows unique to the control condition, and 2, 208 flows unique to the stimulated condition.

Of these 6, 886 implied intercellular flow relations, i.e., the inflow-to-outflow pairs, CellChat gives no information about 1) how many of these pairs are actually statistically dependent and thus represent true signal inflow-to-outflow relationships, and 2) of the pairs that actually are statistically dependent, what are the intracellular mechanisms that mediate these intercellular flows. FlowSig builds on top of analysis provided by methods such as CellChat by combining cell-cell communication output, which describe intercellular patterns, with gene expression modules inferred by methods such as pyLIGER, which describe coordinated intracellular gene expression patterns, and thus is able answer these questions and connect how inflows of intercellular signals are processed via specific intracellular mechanisms, i.e., transcription factors, which then are converted into outflows of other intercellular signals. In fact, FlowSig only infers that 44 out of the preliminary 6, 886 inflow-outflow pairs are actually statistically dependent.

- Continuing on baselines: The quality of insight "FlowSig identified that FGF, IL6, MDK, and SST were the dominant drivers of intercellular flows " could also be derived from a cell communication model of pairs of cell types like CellChat, so also here, demonstrating how this insight is reproduced / improved would make this more convincing.

Thank you for this comment. In the revised Results section, on page 17, when presenting the results in Figure 4, we expanded on the insights that FlowSig identifies, in particular those that cannot be identified from CellChat analysis alone.

Specifically, we point out that of the 35 signaling pathways that CellChat inferred to be active and of the 3, 987 suggested intercellular flows from direct CellChat output, FlowSig identifies that only inflow through FGF (via FGFR1), IL6 (via IL6R + IL6ST), MDK (via NCL), and SST (SSTR2) actually lead to secondary signal outflow via GCG, INHBA, NAMPT, SPP1, TGFB1, TNFSF12, and UCN3. Furthermore, by extracting the top contributing TFs to GEM-1, GEM-3, GEM-4, GEM-5, and GEM-6, we observed that FGF, IL6, MDK, and SST mediates signal outflow through the transcription factors ID1, NR1D1, TFF3, and ZNF419. These analyses and conclusions could not be drawn from neither CellChat output alone nor pyLIGER output alone, and could only be inferred after integrating the multiplex cell-cell communication networks with intracellular gene expression modules.

- Similarly, it would be interesting to discuss further how this method adds value over CellChat or other cell communication models in the COVID example, ie in the case where communication

patterns are not stimulated, and thus, harder to interpret than in the previous "validation" examples in the manuscript. For example, the result "In this case, FlowSig found that increasing severity of COVID-19 is associated with 1) a gradual loss of regulatory intercellular inflows and 2) an increase of inflammatory chemokine outflow that is driven by macrophages and neutrophils." could be contrasted with a baseline. One potential issue that I see is that the flow modules carry more information than pairwise associations (good!), but may also be more complicated to interpret. This could be addressed by showcasing how this new method yield 1) new results and 2) usable results among the new results, specifically in this example.

Thank you for this suggestion. In the revised Results section, on pages 19–20, when presenting the results in Figure 5, we expanded on the insights that FlowSig identifies for the COVID-19 example, in particular those that cannot be identified from CellChat analysis alone.

We highlight that in the case of the COVID-19 data, the number of inferred active signaling pathways for both Moderate and Severe COVID-19—55 and 54, respectively—is significantly greater than the number of pathways inferred for Healthy Controls (46). This analysis alone does not indicate how these signaling pathways may relate to Severe COVID-19. However, we did observe that as COVID-19 severity increases, the number of inferred regulatory signal inflows and GEMs decreases.

We observe that, with respect to Healthy Controls, in Moderate COVID-19, signal inflow via AXL, CD4, F2RL1, ITGAX+ITGB2, TNFRSF12A and TNFRSF14 no longer drive intercellular flows but now inflow through CAP1 does. With respect to Moderate COVID-19, in Severe COVID-19, signal inflow via CD27, CXCR3, FPR1, IL6R+IL6ST, LTBR, NCL, NRP2+PLXNA2, SDC1, TNFRSF13B, TNFRSF17, and TNFRSF25 no longer drive intercellular flows, but like in Healthy Controls, inflow via AXL, F2RL1, and, TNFRSF14 drive intercellular flows, as does inflow into CD4.

We also note that between Healthy and Moderate COVID-19, GEM-4, GEM-10, GEM-12, GEM-14 do not mediate signal outflows, while GEM-7 does. We observe that GEM-4 is associated with Epithelial cells, GEM-10 is associated with Plasma cells and T cells, GEM-12 is associated with Macrophages and Neutrophils and Severe COVID-19, while GEM-7 is associated with Mast cells, suggesting that Moderate COVID-19 is associated with an increase in the role of Mast cells but a reduction in role of Epithelial cells and several immune cell types, including Plasma cells, T cells, Macrophages, and Neutrophils. From Moderate to Severe COVID-19, the GEMs GEM-1, GEM-2, GEM-5, GEM-11, GEM-13, GEM-18, and GEM-19 no longer mediate signal outflows, but now gain of GEM-12 and GEM-14 do. We observe that GEM-1, which is associated with Macrophages and mDC cells, GEM-2 is associated with B, Plasma, and pDC cells, GEM-5 is associated with Epithelial cells, GEM-11 is T cells, GEM-13 is associated with B, NK, and T cells, GEM-18 is associated with Epithelial cells, and GEM-19 is enriched for all conditions and cell types, while GEM-12 is associated with Macrophages and Neutrophils. This

suggests that severity of COVID-19 is linked to the interplay between different cell types, the signals they secrete and their regulation of such signals.

- Glancing over the Github repo, the code seems well maintained.

Thank you.

Clarity and context: lucidity of abstract/summary, appropriateness of abstract, introduction and conclusions:

- I was not able to access the code under the described URL

"<https://github.com/axelalmet/cellflowsig>" but found "<https://github.com/axelalmet/flowsig>" which I assume is this work.

Thank you for pointing this discrepancy. We have fixed the link in the updated manuscript in the Methods section.

Decision Letter, first revision:

Our ref: NMETH-A53442A

16th May 2024

Dear Dr. Nie,

Thank you for submitting your revised manuscript "Inferring pattern-driving intercellular flows from single-cell and spatial transcriptomics" (NMETH-A53442A). It has now been seen by the original referees and their comments are below. The reviewers find that the paper has improved in revision, and therefore we'll be happy in principle to publish it in Nature Methods, pending minor revisions to satisfy the referees' final requests and to comply with our editorial and formatting guidelines.

When submitting the revised version, please also supply a point-by-point response letter.

TRANSPARENT PEER REVIEW

Please note: we allow redactions to authors' rebuttal and reviewer comments in the interest of confidentiality. If you are concerned about the release of confidential data, please let us know specifically what information you would like to have removed. Please note that we cannot incorporate redactions for any other reasons. Reviewer names will be published in the peer review files if the reviewer signed the comments to authors, or if reviewers explicitly agree to release their name. For more information, please refer to our FAQ page.

ORCID

Sincerely,

Lin Tang, PhD
Senior Editor
Nature Methods

Reviewer #1 (Remarks to the Author):

The authors have addressed all of my comments, and I believe that the manuscript has improved substantially.

My only comment is that it seems that the two python notebook tutorials on github are not accessible.

Reviewer #2 (Remarks to the Author):

Summary of the key changes:

- The authors provided further analyses that dissect the performance of FlowSig with respect to

modules in the preprocessing pipeline, where they added further GEM and cell-cell communication modules. The insight on converging results with respect to core transcription factors is interesting.

- The authors provided a comparison with key multicellular representation learning methods. The comparison is qualitative, but we acknowledge that it is difficult to provide meaningful quantitative benchmarks in this setting.

Remaining comments:

- It would be useful to consolidate both text and figures to highlight key results and comparisons. Currently, a lot of information is packed into the main text and figures that may distract readers from the key insights.

Minor comments

- L47 "These two types of patterns are often treated as independent when, in reality, significant interplay between both ligand-receptor interactions and GEMs drives intercellular flows across tissues." We suggest toning this down, many studies now try to interpret GEMs in terms of ligand/receptor activity.
- L52 You could consider adding a supplementary table with model features that you find meaningful to distinguish this work from others, that could help make this paragraph more accessible.
- L73: "When analyzing non-spatial scRNA-seq data, where true ligand-receptor interactions are harder to infer, we incorporate information perturbation data by performing conditional invariance testing to infer a more accurate CPDAG." Information perturbation is unclear at this point in the manuscript.
- Fig. 4d,e,f are hard to visually parse, the authors could try to restructure the aspect ratio of these panels for example and avoid arrows overlaying text. The same goes for multiple other instances of this type of plot.

Reviewer #2 (Remarks on code availability):

The code is packaged and documented.

Author Rebuttal, first revision:

Response to Reviewer #1

Reviewer #1:

The authors have addressed all of my comments, and I believe that the manuscript has improved substantially.

We thank Reviewer for this positive comment.

My only comment is that it seems that the two python notebook tutorials on github are not accessible.

We thank Reviewer for this suggestion and apologize that the Jupyter notebook tutorials were not completely up to date.

We have now updated both notebooks that demonstrate how to apply FlowSig to non-spatial scRNA-seq and spatial Stereo-seq data at <https://github.com/axelalmet/flowsig/blob/main/flowsig/tutorials>. These notebooks can also be viewed on one's local machine by cloning the GitHub repository.

Response to Reviewer #2

Reviewer #2:

Summary of the key changes:
- The authors provided further analyses that dissect the performance of FlowSig with respect to modules in the preprocessing pipeline, where they added further GEM and cell-cell communication modules. The insight on converging results with respect to core transcription factors is interesting.
- The authors provided a comparison with key multicellular representation learning methods. The comparison is qualitative, but we acknowledge that it is difficult to provide meaningful quantitative benchmarks in this setting.

We thank Reviewer for these positive comments and agree with the summary.

Remaining comments:
- It would be useful to consolidate both text and figures to highlight key results and comparisons. Currently, a lot of information is packed into the main text and figures that may distract readers from the key insights.

We thank Reviewer for this useful suggestion and have updated the manuscript to better consolidate text and figures.

Minor

comments

- L47 "These two types of patterns are often treated as independent when, in reality, significant interplay between both ligand-receptor interactions and GEMs drives intercellular flows across tissues." We suggest toning this down, many studies now try to interpret GEMs in terms of ligand/receptor activity.

We agree with Reviewer and have updated the Introduction to tone down this language.

- L52 You could consider adding a supplementary table with model features that you find meaningful to distinguish this work from others, that could help make this paragraph more accessible.

We thank Reviewer for this helpful comment and have updated the Supplementary Information to include what is now Supplementary Table 1, which summarizes the similarities and differences between FlowSig and other methods to which we compared FlowSig.

- L73: "When analyzing non-spatial scRNA-seq data, where true ligand-receptor interactions are harder to infer, we incorporate information perturbation data by performing conditional invariance testing to infer a more accurate CPDAG." Information perturbation is unclear at this point in the manuscript.

We thank the Reviewer for pointing out this ambiguous statement. We have updated the Introduction to clarify better what we mean by "information perturbation data". Namely, that we incorporate additional datasets of the same population that have been perturbed from the "control" condition via external stimulation, disease, age, time, or any biologically meaningful axis.

- Fig. 4d,e,f are hard to visually parse, the authors could try to restructure the aspect ratio of these panels for example and avoid arrows overlaying text. The same goes for multiple other instances of this type of plot.

We thank the Reviewer for this suggestion and have updated Fig. 4d–f and other similar plots in Fig. 3d–e, Fig. 5d–f, and Fig. 6b–c, by changing their aspect ratios and moving text labels so that they do not cover flow network arrows.

Final Decision Letter:

23rd Jul 2024

Dear Dr Nie,

I am pleased to inform you that your Article, "Inferring pattern-driving intercellular flows from single-cell and spatial transcriptomics", has now been accepted for publication in Nature Methods. The received and accepted dates will be 4th Aug 2023 and 23rd Jul 2024. This note is intended to let you know what to expect from us over the next month or so, and to let you know where to address any further questions.

Over the next few weeks, your paper will be copyedited to ensure that it conforms to Nature Methods style. Once your paper is typeset, you will receive an email with a link to choose the appropriate publishing options for your paper and our Author Services team will be in touch regarding any additional information that may be required. It is extremely important that you let us know now whether you will be difficult to contact over the next month. If this is the case, we ask that you send us the contact information (email, phone and fax) of someone who will be able to check the proofs and deal with any last-minute problems.

Please note that *Nature Methods* is a Transformative Journal (TJ). Authors may publish their research with us through the traditional subscription access route or make their paper immediately open access through payment of an article-processing charge (APC). Authors will not be required to make a final decision about access to their article until it has been accepted. Find out more about Transformative Journals

Please feel free to contact me if you have questions about any of these points. Thank you very much for publishing your work at Nature Methods!

Best regards,

Lin Tang, PhD
Senior Editor
Nature Methods